# High-fidelity aeroelastic analyses of wind turbines in complex terrain: Fluid-Structure Interaction and aerodynamic modelling

Giorgia Guma[1], Philipp Bucher[2], Patrick Letzgus[1], Thorsten Lutz[1], and Roland Wüchner[3]

[1]Institute of Aerodynamics and Gas Dynamics, University of Stuttgart, Pfaffenwaldring 21, 70569 Stuttgart, Germany
[2]Chair of Structural Analysis, Technical University of Munich, Arcisstr. 21, 80333 Munich, Germany
[3]Institute of Structural Analysis, Technische Universität Braunschweig, Beethovenstr. 51, 38106 Braunschweig, Germany

**Correspondence:** Giorgia Guma (guma@iag.uni-stuttgart.de)

**Abstract.** This paper shows high-fidelity Fluid Structure Interaction (FSI) studies applied on the research wind turbine of the WINSENT project. In this project, two research wind turbines are going to be erected in the South of Germany in the WindForS complex terrain test field. The FSI is obtained by coupling the CFD URANS/DES code FLOWer and the multiphysics FEM solver Kratos, in which both beam and shell structural elements can be chosen to model the turbine. The two codes are coupled in both an explicit and an implicit way. The different modelling approaches strongly differ with respect to computational resources and therefore the advantages of their higher accuracy must be correlated with the respective additional computational costs. The presented FSI coupling method has been applied firstly to a single blade model of the turbine under standard uniform inflow conditions. It could be concluded that for such a small turbine, in uniform conditions a beam model is sufficient to correctly build the blade deformations. Afterwards, the aerodynamic complexity has been increased considering the full turbine with turbulent inflow conditions generated from real field data, in both a flat and complex terrains. It is shown that in these cases a higher structural fidelity is necessary. The effects of aeroelasticity are then shown on the phase-averaged blade loads, showing that using the same inflow turbulence, a flat terrain is mostly influenced by the shear, while the complex terrain is mostly affected by low velocity structures generated by the forest. Finally, the impact of aeroelasticity and turbulence on the Damage Equivalent Loading (DEL) is discussed, showing that flexibility is reducing the DEL in case of turbulent inflow, acting as a damper breaking larger cycles into smaller ones.

## 1 Introduction

According to the "Renewable Energy Statistics 2020" of the International Renewable Energy Agency (IRENA), the global installed wind power capacity increased by a factor of almost 83 in the last twenty years, from around 7.5 GW in 1997 to around 622 GW in 2019. In 2018 wind energy represented around 19% of the total electricity produced by renewables worldwide. This makes wind energy the most growing renewable power technology nowadays. One of the reasons for its extreme development is the strong investment in the research of new materials and construction techniques in order to reach larger and lighter rotor designs. According to (IEA), the global average cost of electricity from onshore fell from 76 USD/MWh in 2016 to 53 USD/MWh in 2019, and it is expected to decrease 15% during 2020-2025, expanding the market of bankable projects to low wind speed areas and complex terrains. A complex terrain is a terrain where topology and roughness have a significant impact

on the wind in the Atmospheric Boundary Layer (ABL). For this reason, differently from an offshore wind turbine that is characterized by high inflow velocities and low Turbulence Intensity (TI), a complex terrain has exactly the opposite attributes. This makes it difficult to estimate the wind potential and consequently the performances of the installed wind turbine. Therefore high-fidelity is necessary to simulate the site-specific wind field and turbulence. It needs to be considered that the produced power of a turbine is proportional to the cube of the velocity, and therefore a 3% error in the velocity leads to a 9% error in the power calculation. Those effects can be already mapped using RANS methods as in Brodeur and Masson (2008). On the other side, higher fidelity models such as hybrid RANS/LES are needed to catch the effects on the turbulence of smaller vortices from the ground (Bechmann and Sørensen, 2010). The applicability of Detached Delayed Eddy Simulations (DDES) for wind turbines was shown by Weihing et al. (2018), while DDES investigations of a complex terrain have been conducted by Schulz et al. (2016), focusing on the performance of the turbine. A widely studied complex terrain is the double ridge in Perdigão in Portugal, where a single turbine has been erected with consequent measurements campaigns in 2017 (Fernando et al., 2019). On the other side, the increase of rotor sizes with the consequent necessity of slender blades makes it impossible to neglect aeroelastic effects, and that is why aeroelastic models based on different fidelity levels are widely developed within the research community. DTU, for example, widely uses the Fluid Structure Interaction (FSI) coupled developed by Heinz et al. (2013) and known as HAWC2CFD between the CFD solver EllipSys3D (Sørensen, 1995; Michelsen, 1992) and the finite-element multibody serial solver HAWC2 (Larsen and Hansen, 2007). Heinz et al. (2016a) used it on the NREL 5 MW rotor and compared it to Blade Element Momentum (BEM) based calculations using HAWC2, too. Most discrepancies were shown when the turbine was in standstill, although good agreement was found in uniform and yawed conditions. Li et al. (2017) used the same turbine adding a turbulent inflow synthetically generated by the use of a Mann box, a multibody drivetrain and a control system. He showed that an active pitching control lead to a more uniformly distributed wake even in turbulent inflow conditions, in comparison to a stall regulated study. The same coupling has been used in Heinz et al. (2016b) to perform studies on Vortex Induced Vibrations (VIV). In this case a single blade model of the DTU 10 MW wind turbine has been take into account finding that the conditions triggering VIV are an Angle Of Attack (AOA) of around $90°$ and a flow inclination between $20°$ and $55°$. Horcas et al. (2020) continued his work analyzing for the same turbine the effect of different tip configurations on the VIV phenomenon. Recently, the HAWC2CFD FSI coupling has been used by Grinderslev et al. (2021) to aeroelastically investigate the 2.3 MW DANAERO rotor using a new turbulence model, combining the Deardorff large eddy simulation (LES) model for atmospheric flow with improved detached delayed eddy simulation (IDDES) model for the near rotor area. The authors found that for such stiff turbines, flexibility has only a marginal effect, while the loading was strongly affected by the inflow turbulence, underlining the importance of its modelling.

Santo et al. (2020a) used a FSI coupling between a CFD URANS model in Ansys and a structural shell model in Abaqus to analyze the effects of tilt, yaw, tower shadow and wind shear. The authors found that yaw leads to a lower deflection but a higher yaw-moment on the hub. In Santo et al. (2020b), gusts were introduced showing that in the considered case, flow separation was occurring working as a passive load control.

Streiner et al. (2008), Meister (2015) and Klein et al. (2018) worked sequentially on a FSI coupling between the CFD code FLOWer and the Multi-Body Simulation (MBS) commercial solver SIMPACK. In Klein et al. (2018), the NREL 5 MW wind

turbine has been simulated, including the drive train torsion, the foundation flexibility and the controller for variation of RPM and pitch angle, examining thereby the origin of low frequency noise sources and seismic excitation. The same coupling has been then used in Guma et al. (2021) to simulate the DANAERO wind turbine in both uniform and turbulent conditions, these ones synthetically generated by the use of a Mann box (Mann, 1994). Results in uniform inflow were compared to a consistent low-fidelity model and analysis on the effects on the Damage Equivalent Loading (DEL) have been carried out. Here it was demonstrated that for the specific inflow case, the deformations have only a marginal influence on the DEL, that shows to be more affected by the inflow turbulence fluctuations. The same CFD solver was coupled by Sayed et al. (2016) to the CSD solver Carat++ and applied to a only blade model of the DTU 10 MW generic rotor under uniform inflow conditions. Both beam and shell elements have been applied showing that an evident error is made when geometric non-linearities are not taken into account.

Dose et al. (2018) coupled the flow solver OpenFOAM to the FEM-based beam solver BeamFOAM to perform simulations of the NREL 5 MW rotor, showing that aeroelasticity is particularly important when a yaw misalignment is taken into account. The same rotor was adopted by Yu and Kwon (2014) using a loose CFD-CSD coupling with an incompressible CFD solver and non-linear Euler-Bernoulli beam elements for the structure. The communication in this case was only once per revolution. The same turbine was also used by Bazilevs et al. (2011) and Bazilevs et al. (2012) by means of FSI between a low-order Arbitrary Lagrangian-Eulerian Variational Multi Scale (ALE-VMS) flow solver and a Non-Uniform Rational Basis Spline (NURBS) based structural solver. An Isogeometric analysis (IGA) has been used in this case, that integrates FEM in the CAD tool so that no geometric approximation is needed. This increases its accuracy and simplicity, especially for form optimization chains.

Within the scope of the present study, a highly accurate CFD-FEM coupling has been built and a model has been created for a small research wind turbine to be erected in a complex terrain location in the South of Germany. Two different structural models and coupling algorithms have been used to determine their impact in different inflow configurations. Starting from a single blade model and a full model of the turbine in uniform standard conditions, up to the turbulent inflow conditions in both flat and complex terrain, the difference in deformations, wake shape, loads and fatigue are analyzed. Considering the strong variation in computational costs within the different configurations, it is of particular interest to determine the respective fidelity requirements. Sect. 2 describes the methodology, from the CFD to the CSD models and the construction of the coupling. Sect. 3 shows the results firstly for a single blade model, then for the full turbine focusing on the wake shape, the impact of the terrain and effects on the fatigue loading.

## 2 Methodology

### 2.1 WINSENT research wind turbine

The project WINSENT (Wind Energy Science and Engineering in Complex Terrain) is a German project (WindForS, 2016) involving different universities and research institutions in the South of Germany. The site is located in Stöttener Berg, a complex terrain location in the Swabian Alb (Fig. 1), where two research wind turbines will be erected. These will be pitch-regulated, and will have around 50 m diameter, 70 m hub height, a tilt angle of 4°, a cone angle of 2.5° and a rated power of

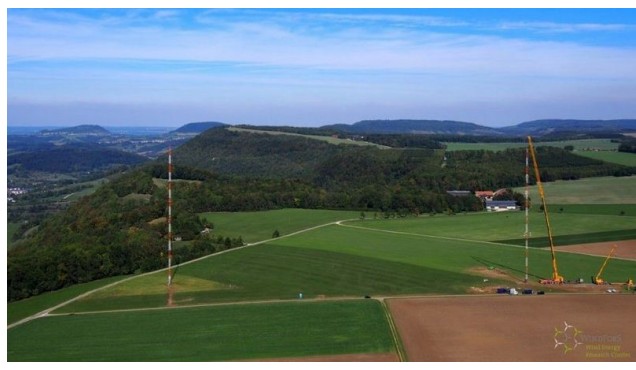

**Figure 1.** WindForS test field in Stöttener Berg (from Testfield) with focus on the two met masts.

around 750 kW, by a rated inflow velocity of 11 m/s and a rotor speed of 26.5 RPM. They will be equipped with measuring technique for the validation of both high (presented in this work) and low fidelity. A computational chain has been developed
within the project, starting from mesoscale simulations up to high-fidelity FSI calculations in the near field around the turbine. A large variety of measurement equipment has been used to characterize the local site-specific wind field, such as unmanned aerial systems (UAS), lidar instruments and met masts with ultrasonic anemometers. This allowed to validate CFD terrain calculations with the data from the met masts as prescribed in Letzgus et al. (2020).

    This paper focuses on the final part of the simulation chain, in which the CFD terrain calculations are used as inflow
conditions to analyze the aeroelastic effects on the research wind turbine by means of high-fidelity FSI.

    Starting from the provided CAD file and material properties, models with different degree of fidelity have been built.

### 2.1.1   CFD model and inflow conditions

The CFD code used within this study is FLOWer (Raddatz, 2009). This was originally developed at the German Aerospace Centre (DLR) and since many years it is expanded at the Institute of Aerodynamics and Gas Dynamics for helicopters and
wind turbines purposes. It is a finite volume URANS and DES code, using structured meshes. Both a second order central cell centered (Jameson et al., 1981) and fifth order weighted essentially non-oscillatory (Kowarsch et al., 2013) spatial discretization schemes are available. The second one is utilized in the background meshes in this study to reduce the turbulence dissipation. A dual-time stepping integration scheme is applied, in which the number of inner iteration of a timestep $t$ is depending on how close a guessed solution is to the final value, based on a prescribed tolerance. An artificial 5-stage Runge-Kutta scheme is used
as time-stepping scheme. Up to three level of multigrid can be utilized to accelerate convergence, although a complex terrain mesh is not suited for it. Single meshes for each component need to be independently generated and then combined by means of the Chimera technique. The Shear-Stress-Transport (SST) k-omega model according to Menter (Menter, 1993) with a fully turbulent boundary layer is used to for the simulations in this study. URANS has been here used for all cases with uniform inflow conditions, while DDES has been activated when turbulence was involved. In this way the turbulence propagation is
resolved by LES, while the areas close to the wall and therefore the boundary layer are resolved by URANS. Bangga et al.

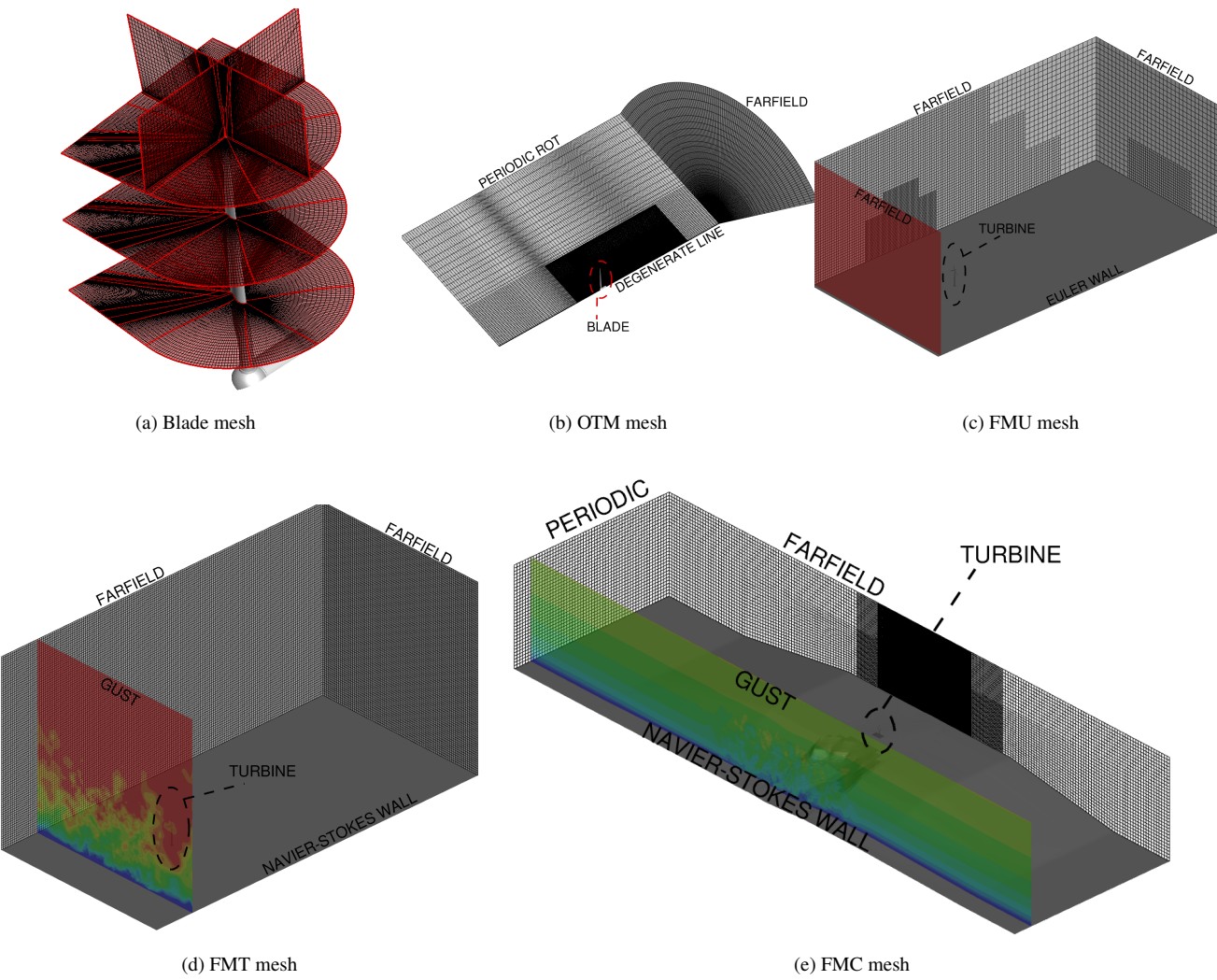

(a) Blade mesh

(b) OTM mesh

(c) FMU mesh

(d) FMT mesh

(e) FMC mesh

**Figure 2.** Details of the meshes

(2017) showed DDES combined to this turbulence model showed the most stable and best results compared to measured data. The CFD model of the turbine has been built starting from the provided CAD files. A "water tight" outer surface has been extracted and meshing has been performed with both use of Pointwise and in-house scripts. All components have been meshed ensuring $y^+ \leq 1$ of the wall nearest cell. A mesh refinement study of the blade has been done in (Guma et al., 2018), (Schäffler, 2019). According to these studies, different mesh properties, characteristics and sizes have been tested to optimize the profile, the trailing and leading edge resolutions, as well as the wake and pinion areas. Different timesteps and number of iterations have been also tested, although always at rated conditions. For this reason the second cheapest number of cells has been then chosen, in order to remain conservative and in line with other similar works performed with the same code. The blade mesh in Fig. 2a has been generated with the following properties:

| Case name | Blade | Hub + Nacelle | Tower | Background | Total |
|-----------|-------|---------------|-------|------------|-------|
| One Third Model (OTM) | 1 x 11 | 2 | - | 18 | 24 |
| Full Model Uniform (FMU) | 3 x 11 | 5.5 | 3 | 40 | 85 |
| Full Model Turbulent (FMT) | 3 x 11 | 5.5 | 3 | 105 | 148 |
| Full Model Complex (FMC) | 3 x 11 | 5.5 | 3 | 300 | 345 |

**Table 1.** Mesh sizes in Mio.

| Case name | Flow Direction | Side Direction | Height |
|-----------|----------------|----------------|--------|
| OTM | -3D;+3.5D | 120° | 3D |
| FMU | -7.5D;+19.5 | ±7D | 10D |
| FMT | -7.5D;+18.5D | ±6.5D | 13D |
| FMC | -22.5D;+10D | ±65D | 46D |

**Table 2.** Background dimensions in rotor diameters from the tower bottom. Values are approximated and referred to the tower center.

– CH mesh;

– 193 sections over the blade with 257 points over the each profile;

– 32 cells in the boundary layer with a growth rate of 1.11;

– a total of around 12 million cells for each blade;

Three different CFD models of the turbine have been built with the following characteristics and abbreviations:

– a one-third model with only one blade (OTM) and uniform inflow conditions (Fig. 2b);

– a full model of the turbine with a flat terrain with uniform (FMU) (fig 2c) or turbulent inflow conditions (FMT) (Fig. 2d);

– a full model of the turbine with a complex terrain (FMC) and turbulent inflow conditions (Fig. 2e);

Thanks to the Chimera technique, the turbine CFD mesh is always the same for the different models, changing only the
background grids.

Table 1 gives a breakdown of the mesh sizes in number of cells for the different setups (that consequently influence the costs of the calculations). The different boundary conditions and CFD models are depicted in Fig. 2. Additionally in Table 2 the dimensions of the different backgrounds are shown in terms of distances from the tower bottom.The complex terrain mesh is clearly much wider than longer, and this is done to avoid any influence of the interaction between the periodic boundary
condition and the topography, as explain in Letzgus et al. (2021). The mesh discretization occurs via hanging grid nodes, ensuring in the closes area to the turbine a resolution of 0.25m, increasing up to 2 m for a large area of the domain to ensure a sufficient physical resolution. In the following the meaning of the boundary conditions is clarified:

| Case name | u Hub Height [m/s] | TI [%] | Shear $\alpha$ [°] | Pitch angle [°] | RPM | Terrain |
|-----------|--------------------|--------|---------------------|------------------|------|---------|
| OTM | 11 m/s | 0 | 0 | 2.59 | 26.5 | No |
| FMU | 11 m/s | 0 | 0 | 2.59 | 26.5 | Flat |
| FMT | 16.5 m/s | 9 | 0.2 | 17.46 | 26.5 | Flat |
| FMC | 16.5 m/s | 9 | 0.2 | 17.46 | 26.5 | Complex |

**Table 3.** Setup of the computed cases (effective conditions at the turbine position).

- NAVIER-STOKES WALL represents the ground and surface boundary conditions considering friction;

- EULER-WALL represents the ground boundary conditions with no friction;

- FARFIELD, where either a fixed velocity or a zero extrapolation conditions can be set for the background border;

- PERIODIC & PERIODIC ROT are the symmetrical boundary conditions in axial and rotational direction, respectively;

- GUST is the FLOWer boundary condition to use when a wind profile and turbulent fluctuations from experiments (or synthetic turbulence) have to be injected in the flow;

The flat terrain for FMU and FMT is different, because in FMU no wind profile and no inflow turbulence are considered,
and therefore it was possible to use a smaller background and increase the cell size, saving in this way computational time. Table 3 summarizes the operating conditions at which the simulations are run with the corresponding RPM and pitch angles of the turbine. No controller is considered in the simulations, i.e. both RPM and pitch angle are kept constant. OTM and FMU represent the rated conditions of the blade, while FMT and FMC are based on sheared turbulent inflow conditions extracted from real wind measurements on the 31st March 2019 at 11 p.m.

The chosen timestep is the same for all coupled simulations and is related to one azimuthal degree. This has been chosen to be stricter than for the only stiff simulation, that for example, in the complex terrain case was set to two azimuthal degrees. On the other side, Sayed et al. (2016) showed that for the previous version of the her used coupling, a one azimuthal degree timestep is necessary. The number of sub iterations needs to be adapted according the coupling type, and this will be addressed in the following sections. The turbulence is injected in terms of fluctuations that are superimposed to a sheared uniform inflow.
The fluctuations have been extracted from experimental time series at the WINSENT test field from already installed met masts. The spectra and standard deviations of all velocity components of the different measurement positions have been extracted and synthetic turbulence was then generated with the Mann model. The main focus was to match the measured turbulence as good as possible, especially at the hub height. Important was then to make the cases with flat and with complex terrain as consistent as possible. That is why, while the turbulence fluctuations were kept the same, the wind shear coefficient and the mean velocity
had to be accordingly changed to keep after the slope the the same shear and hub height velocity for both cases at the turbine position. This is necessary because the hill of the complex terrain has an acceleration effect on the flow and therefore a lower reference velocity has to be taken into account upstream of the hill, in order to get the same mean hub velocity between flat and

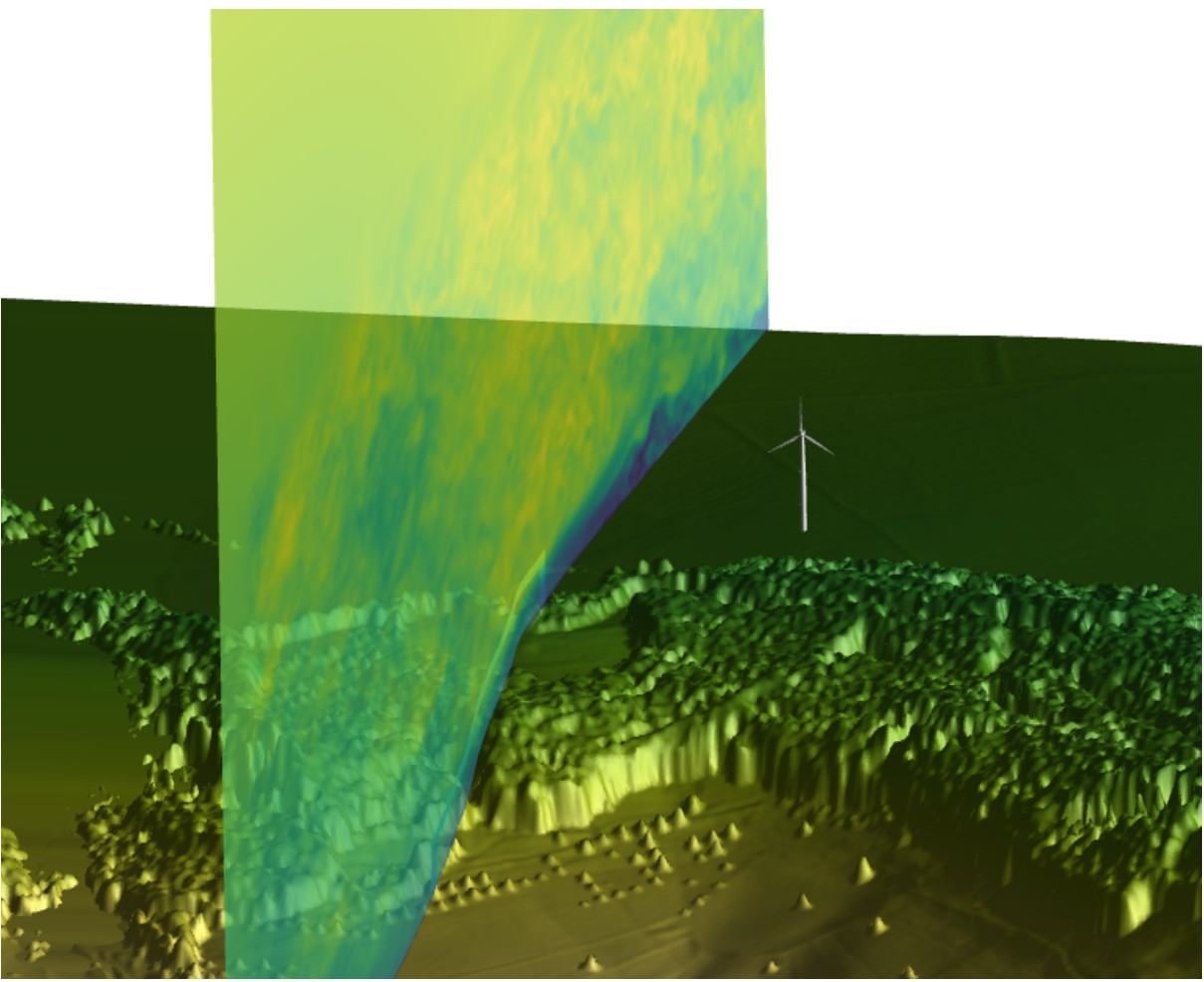

**Figure 3.** CFD model of the complex terrain considering the forest with a slice of the velocity field 2.5 diameters from the turbine on its right hand side.

complex terrain. In the observed time period the atmosphere was neutrally stratified.More information on how the turbulence is injected inside the computational domain is given in Letzgus et al. (2021). The CFD model of the complex terrain is depicted in
Fig. 3, where also the presence of the forest can be noticed. The forest is modeled as additional grid structures by the use of the Chimera technique, in which force terms are added depending on the foliage density and tree height, as described in Letzgus et al. (2020). In this way the modeling can be really easily adjusted to the seasonal conditions and the form of the forest.

| Number of nodes | | |
|---|---|---|
| Coupled Structure | Beam | Shell |
| Blade 1 (2 or 3) | 133 | 13824 |
| Hub | 1 | 1 |
| Nacelle | 1 | 7361 |
| Tower | 36 | 7967 |

**Table 4.** Number of nodes/elements for each structure in both beam and shell models.

| f [Hz] | Manufacturer | Beam | Shell |
|---|---|---|---|
| 1$^{st}$ flap | 1.3 | 1.34 | 1.38 |
| 1$^{st}$ edge | 2.2 | 2.25 | 2.18 |
| 2$^{nd}$ flap | 3.8 | 3.92 | 3.99 |
| 2$^{nd}$ edge | 6.8 | 6.98 | 6.53 |

**Table 5.** Comparison of the first 4 eigenfrequencies of the blade.

### 2.1.2 CSD models

The CSD code applied within this study is the solver Kratos Multiphysics (Dadvand et al., 2010).Kratos was utilized to both
generate the structural models and calculate the structural response. Additionally, it served as coupling interface between the
solvers.

Two structural models have been created for the turbine, one with beam and one with shell elements, respectively. Both
models take into account geometric non-linearities and nonlinear dynamic analysis is applied to solve the turbine structural
model. The main physical differences between both models is that in a beam model neglects the cross section aerodynamic
shape deformation. Secondly, the beam theory implemented into Kratos does not consider bend-twist coupling, while the shell
theory contemplates it intrinsically. Rayleigh damping is used to model the damping properties of the turbine, adapting the
coefficients to each single coupled structure.

From a practical point of a view, a beam and a shell model strongly differ on the number of computational nodes, as shown
in table 4, in which it can be noticed that the hub has been modeled in both cases as an only point. This influences directly the
computational time. Both structural models are depicted in Fig. 4c.

The shell coupling needs a mapper to interpolate the loads between the CFD and CSD meshes. In this study the Mortar
mapper (Wang, 2016) is chosen in Kratos, as before in Sayed et al. (2016). This is not necessary to map the deformations,
because these are directly communicated to FLOWer which uses them as a cloud of points around the structures, deforming
them according to a Radial-Basis Function (RBF) algorithm.
In Kratos it is possible to evaluate the eigenfrequencies of the structural model. The manufacturer delivered the first four
eigenfrequencies of the blade, that are therefore compared in table 5 to the beam and shell models, respectively.

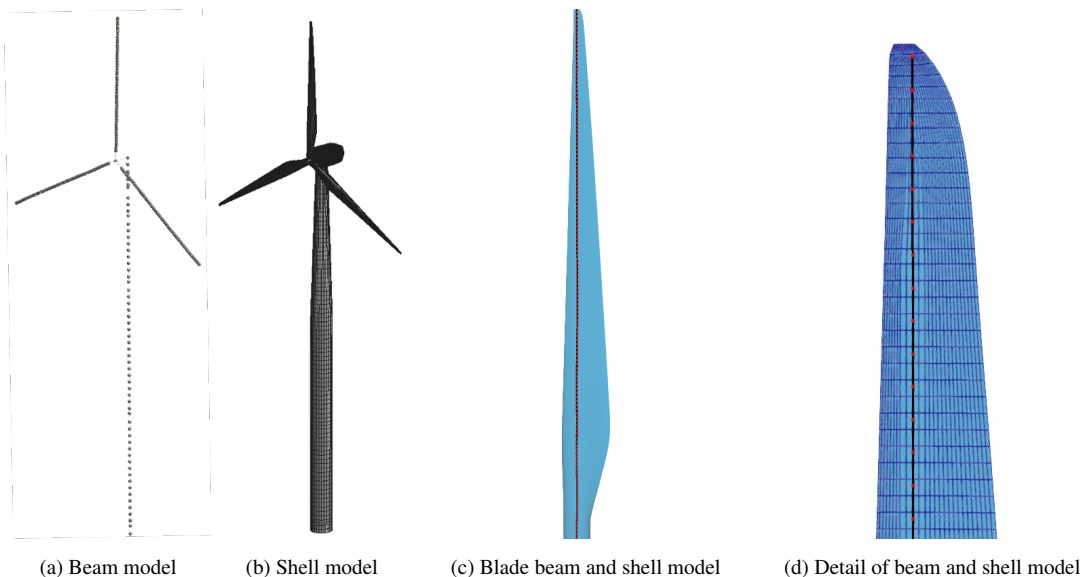

(a) Beam model      (b) Shell model      (c) Blade beam and shell model      (d) Detail of beam and shell model

**Figure 4.** Beam and shell structural models

The entire turbine is modeled in both beam and shell models, in which it is clear from Table 4 that the rotor represents around 90% and 80% of the total amount of cells for beam and shell, respectively. For more information about the structural models used within this study, please refer to Bucher et al..

### 2.1.3   FSI coupling and performances

The FLOWer-Kratos coupling has been developed within the WINSENT project in cooperation with TUM-LST. It is based upon the coupling employed by Sayed et al. (2016) within FLOWer and Carat++. It consists of a partitioned approach with both explicit and implicit algorithms, i.e. the communication between the two solvers happens once (explicit) or several times (implicit) per physical timestep. Both nonlinear beam and shell elements can be coupled and both rotating and non-rotating components can be taken into account, allowing to consider the aeroelasticity of the complete turbine. The communication bases on fileIO and the logic tree is depicted in Fig. 5. The two solvers start independently, and a mesh initialization check takes place at the beginning to ensure a correct mapping of forces and deformations between the CFD and CSD meshes. Kratos is the first solver sending either zero deformations (at the very first coupling timestep) or the last saved deformations. FLOWer deforms the surface and the surrounding meshes according to a Radial-Basis-Function (RBF) algorithm. A direct application of the deformation occurs during the shell coupling, because the deformations are mapped by Kratos directly on the real CFD surface grid. Afterwards, the timestep calculation as prescribed in Sec. 2.1.1 takes place and the resulting aerodynamic forces are calculated. These are either the forces on the three directions in space over all surface cells (in case of shell coupling) or three forces and three moments integrated according to the communication nodes (in case of a beam coupling). Kratos receives the input from FLOWer, maps it on the structural mesh and adds the centrifugal and gravitational forces. The structural timestep

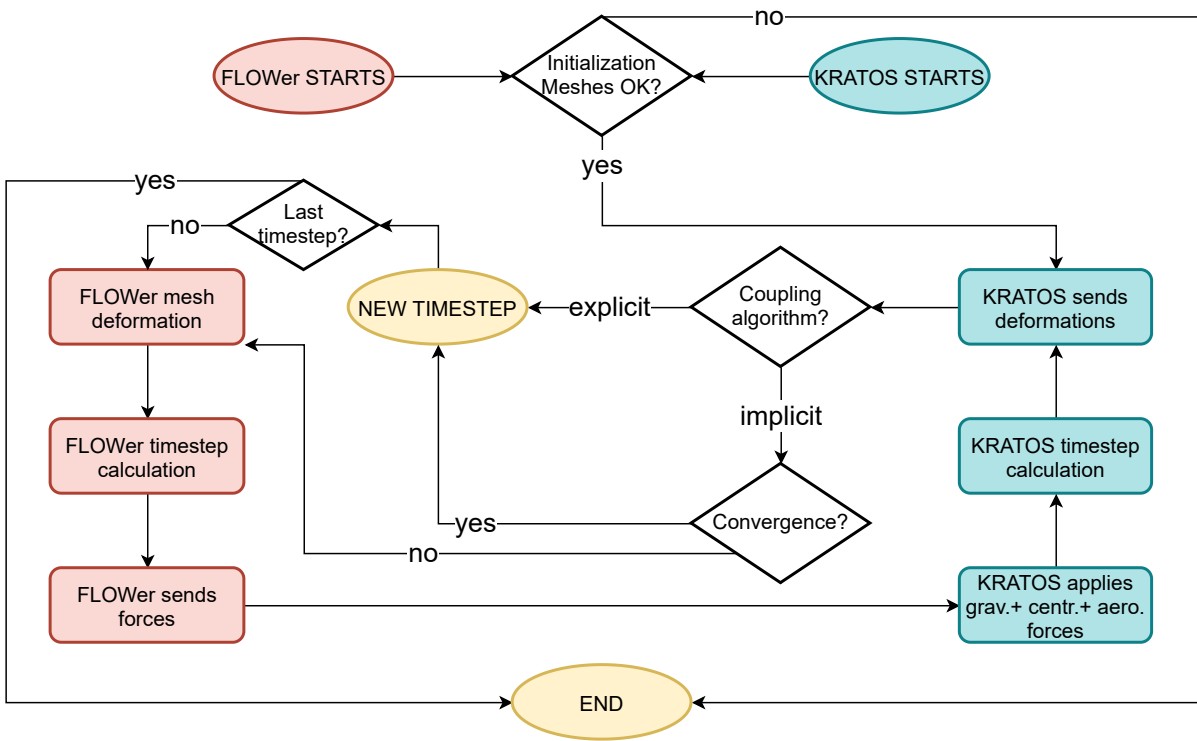

**Figure 5.** Logic tree of the FSI coupling.

is then performed and the resulting deformations are calculated. In case of an explicit coupling, a new physical timestep starts. If the chosen algorithm is implicit, the timestep is repeated until reaching a predefined convergence criteria of the deformations.

One of the main differences between a beam and a shell coupling is the number of Degrees of Freedom (DOF) necessary to fully described the total deformation. A beam element needs 6 DOF (3 translations and 3 rotations) that are calculated in Kratos on prescribed nodes at the shear centers. On the other side, a shell only requires 3 translations because the entire outer

shape of the turbine is taken into account. The choice of the shear centers as communication nodes in the beam coupling is not casual, because a beam model in Kratos is not considering bend-twist coupling. The shear center is by definition the point on which a shear force causes no twist, and a torque moment causes no displacement of the shear center, as described also in Fedorov (2012). A shell model, on the other side, considers intrinsically a bend-twist coupling. As shown in the structural model description in Sec. 2.1.2, a shell model has many more nodes than a beam model, and therefore its computational time

is not negligible anymore as in the case of a beam model. All calculations have been run on the SuperMUC-NG supercomputer at the Leibniz-Rechenzentrum in Munich and an example of computational time for each configuration is given in table 6. As

| CFD / CSD | Rigid | Beam Coupling | Shell Coupling |
|-----------|-------|---------------|----------------|
| Only Blade | 5.28 | 7.92 ($\sim 100\%$ FLOWer) | 13.2 ($\sim 75\%$ FLOWer) |
| Full Turbine | 47.6 | 70.56 ($\sim 100\%$ FLOWer) | 117.6 ($\sim 70\%$ FLOWer) |

**Table 6.** Performance of the FSI coupling based on beam and shell elements in comparison to pure CFD calculations. CPU/h per timestep (one azimuthal degree) in average over one revolution.

it can be seen, FSI computations based on a beam model of the blade/turbine increases the computational time of around 50%. The CSD calculation time is negligible in this case and almost the entire computational time is driven by the CFD part. The increase in computational time is because the number of inner iteration per timestep had to increase in order to reach the same convergence level as only CFD. Shell based FSI calculations have additionally a longer computational time of the CSD solver, due to the high number of structural elements, which results in CFD representing only $\sim 75\%$ of the average timestep. The computational time of the case "Full Turbine" is considered for uniform inflow conditions, which is the cheapest case from the CFD side.

All coupled simulations that will be analyzed in the following sections started from already run stiff cases in order to accelerate the convergence in flexible conditions and reduce the transition times.

## 3 Results

### 3.1 One-third model (OTM) with laminar uniform inflow

A OTM calculation of a wind turbine is a highly efficient way to get a first insight into the turbine performances and, in the case of the development of a new FSI coupling, to test its functioning, capabilities and limitations. The rigid simulation of the OTM case has been firstly run for more than 20 revolutions to ensure the loads convergence and the correct wake development. Afterwards, five revolutions have been computed with flexibility using a timestep equivalent to one azimuthal degree with an explicit algorithm, reaching the complete convergence of the deformations. As described in Sec. 2.1.2, the two structural models are consistent within each other, and the tip deflection differ less than 1%. Linear models lead to a nonphysical elongation of the blade when bending. This drives to large errors in the deformation evaluation, especially when long blades are taken into account. In fact the elongation generates a larger surface of the blade with consequent higher loads and therefore larger bending deformations. This is not happening with the nonlinear models applied in this study, as it can be seen in Fig. 6a.

The tip deformations in flapwise and edgewise direction are depicted in Fig. 6b and 6c. The Normalized Root Mean Square Deviation (NRMSD) is around 1% and therefore negligible. Because the damping is exactly implemented in the same way in both models, this difference is to be associated only to the structural model. The sectional torsion has been calculated from the surface output files, because the shell coupling communicates only the displacements in the three directions. A reference section at the tip of the rigid blade has been taken into consideration, and its indexes have been used to find the same section on the deformed blades. The deformed sections have been afterwards unrotated using the Rodrigues' rotation formula to make

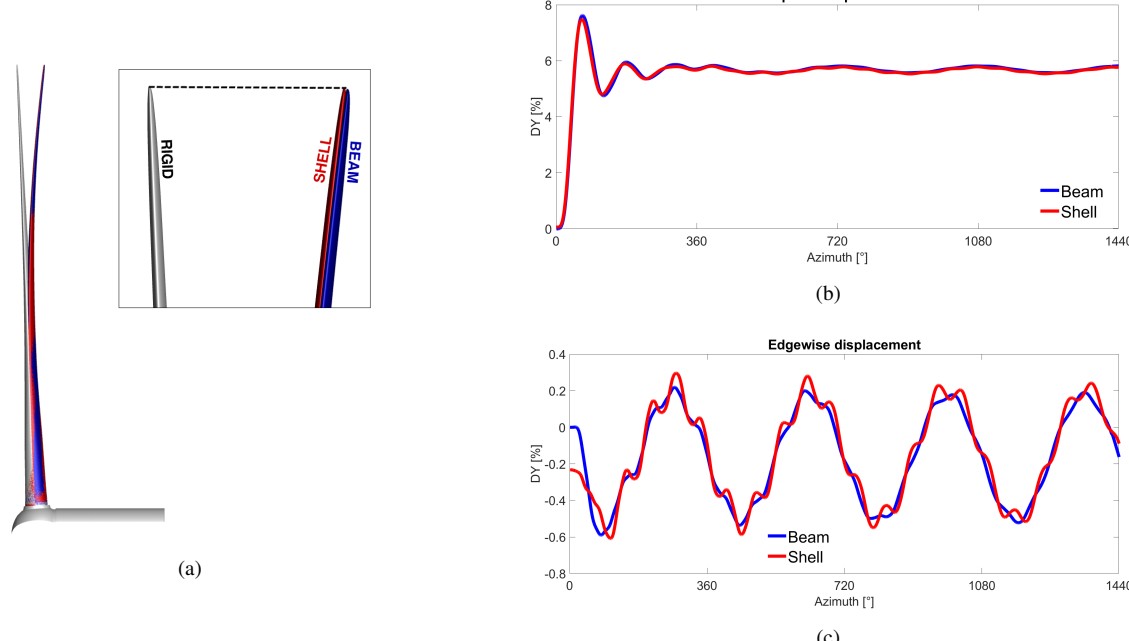

**Figure 6.** One-third model (OTM), rigid and deformed blade beam vs shell. Deformations are normalized according to the blade radius and shown as percentage of the of it.

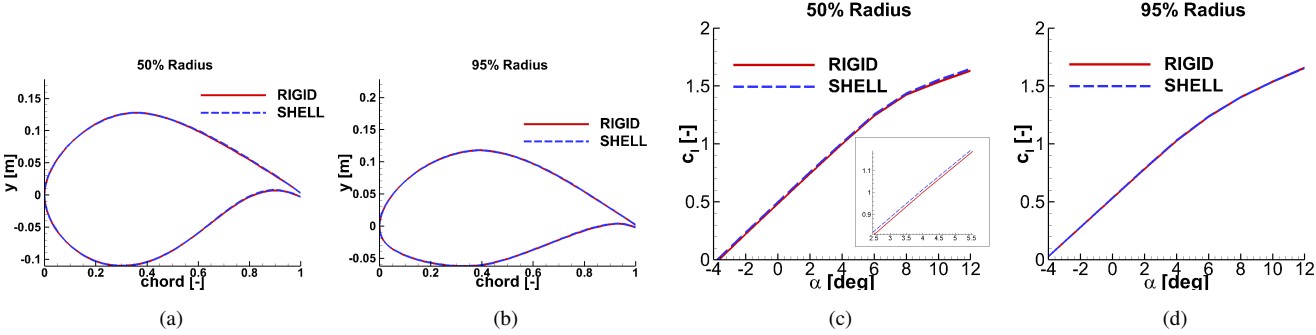

**Figure 7.** Blade sections enlarged in y-direction and $c_l$ calculations for r/R=50% and r/R=95%.

them parallel to the rigid section. Aligning then the leading edges, it is possible to evaluate the torsion angle at the section. The average value of the torsion in case of beam coupling is $0.1138°$, while for the shell model is $0.138°$. The blade is really small and that is why the torsion is negligible. The small difference between beam and shell is to be reconnected to the neglecting of bend-twist coupling in the beam model.

An added value of a shell model is that airfoil deformation can be predicted. Although this can be really limited, noticeable changes in the camber can have an impact in terms of lift and drag coefficients of the airfoil (i.e. the polars). That is why

250

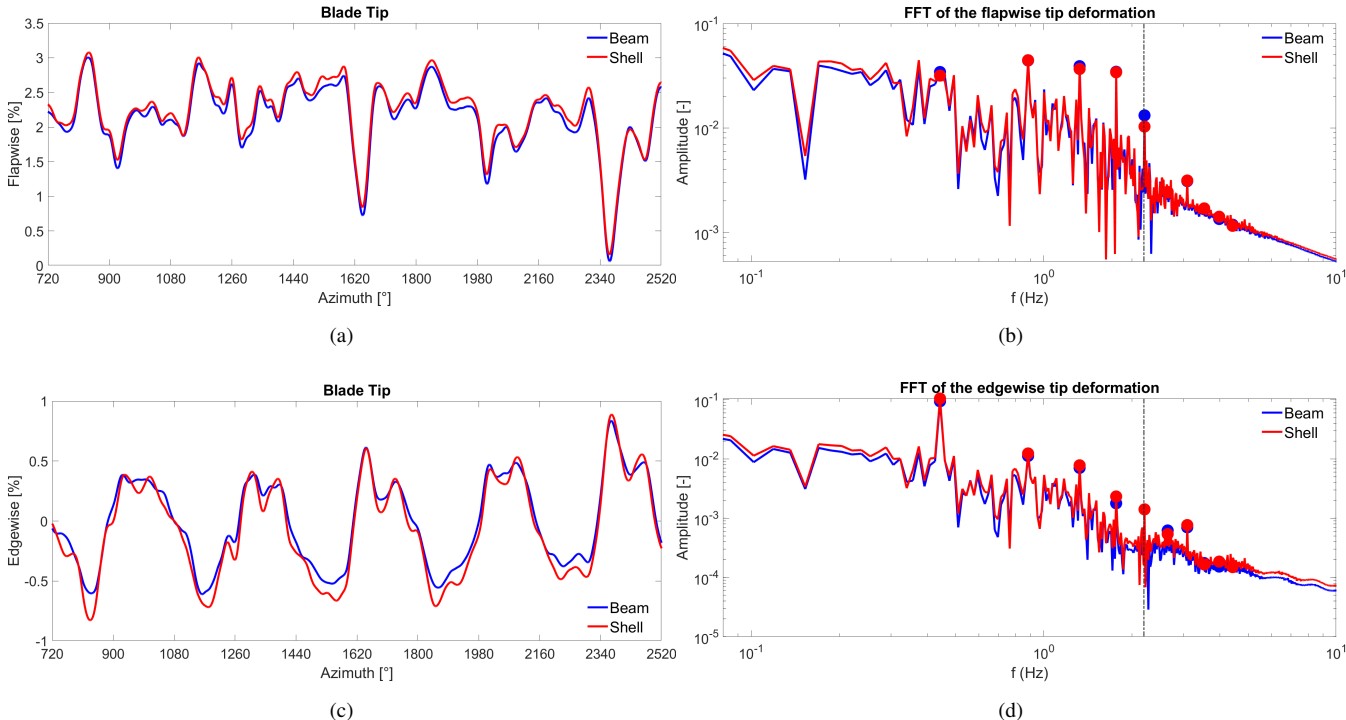

**Figure 8.** Flapwise (a) and edgewise (c) blade tip deformation over three revolutions, comparison of beam and shell coupling. (b) and (d) show the FFT over one minute simulation. A 1-D Moving-Averaging filter has been applied for clearness. Dotted vertical lines are plotted at the first flapwise and edgewise eigenfrequency of the blade, respectively.

2 sections of the deformed and undeformed blade have been extracted, opportunely rotated to make them parallel within each other and analyzed with Xfoil (Drela, 1989) afterwards. Viscous polar calculations have been performed fixing Re=$4e06$ and Mach=$0.19$. The results are depicted in Fig. 7, and no difference could be detected in the outer region, while a light cambering in the middle blade region appeared, leading anyhow to only around $1\%$ difference in the lift coefficient. It can be therefore concluded that for the applied inflow conditions (that are also the standard operating conditions), the usage of only a beam model for this turbine in comparison to a shell model leads to a negligible difference that does not justify the higher computational cost.

### 3.2 Impact of the structural model and coupling algorithm

In this section the use of beam and shell elements is compared for the turbulent inflow case in complex terrain (FMC) in Fig. 8. This has been chosen because it represents the most complex case from the aerodynamic point of view, and therefore of interest for studies about the structural impact. In these cases the entire turbine is considered as flexible.

The choice of the structural elements shows in average a difference of $3.7\%$ of the Root Mean Square (RMS) value of the shell flapwise deformation, which is not negligible anymore in comparison to the OTM case. The difference gets much stronger

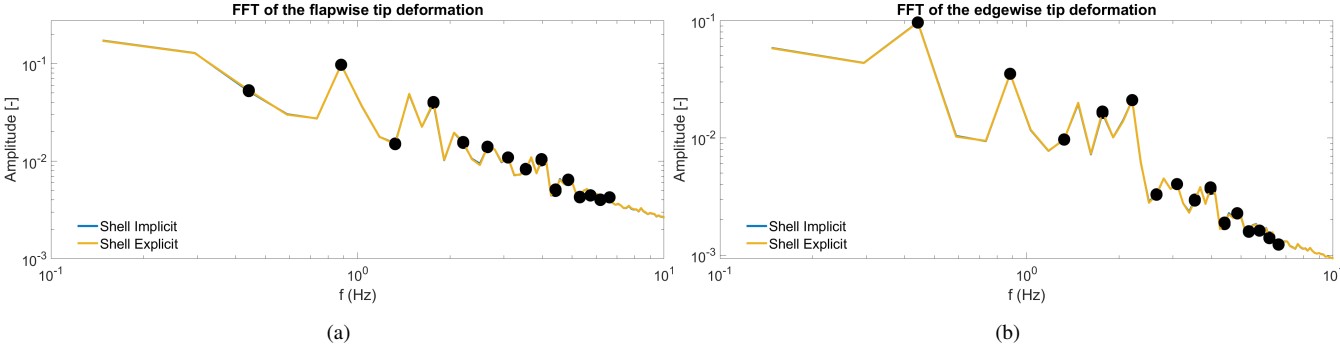

**Figure 9.** FFT of the flapwise and edgewise blade tip deformation using shell elements with an explicit and implicit coupling algorithm. Circle on the RPM harmonics.

in edgewise direction, where an average difference of 15% of the RMS value occurs. The higher frequency oscillations are stronger pronounced in the shell coupling, which can be lead back to the absence of coupling terms in the stiffness matrix of the beam model (bend-bend and bend-twist). On the other side a complete modelling of the geometry is done in the shell model, justifying the differences that are occurring.

The FFT of the signals over one minute are depicted in 8b and 8d. It can be seen that the first harmonic of the rotational velocity (commonly known as 1P) shows clearly the strongest peak in case of the edgewise deformation, suggesting the importance of the weight effect for this deformation direction.On the other side, the 1P frequency does not clearly deviate from the neighboring peaks in flapwise direction, while becoming the strongest peaks only at higher frequencies. This shows that, in this case, the low frequency spectrum of the flapwise deformation is mostly influenced by the turbulent inflow and not the blade-tower passage. The spectrum amplitude in case of the shell coupling is always slightly higher than in beam coupling (especially in edgewise) although the RPM harmonics' amplitude are always close to each other. This has been reconducted again to the absence of some coupling terms in the stiffness matrix of the beam model and the higher accuracy of the shell model in which the entire geometry is taken into account. Only the RPM harmonic close to the edgewise eigenfrequency of the blade shows a higher amplitude in shell than in beam coupling. This suggests that in case of particular excitation of the edgewise eigenfrequency, an aeroelastic instability could take place that the beam coupling would not recognize.

The improvement in accuracy by using an implicit coupling is now discussed. In this case the computational time increases due to the repetition of the timestep. The implicit coupling has only been considered for the FMC case with shell coupling. A total of three revolutions with a timestep equivalent to 1 azimuthal degree have been computed. In order to reach the predefined convergence, it was necessary about 28 times per revolution to perform more coupling iterations per timestep. A FFT of the flapwise and edgewise tip deformation is depicted in Fig. 9. Almost no difference can be seen between the two algorithms in this application, and that is why all following studies have been based on an only explicit algorithm.

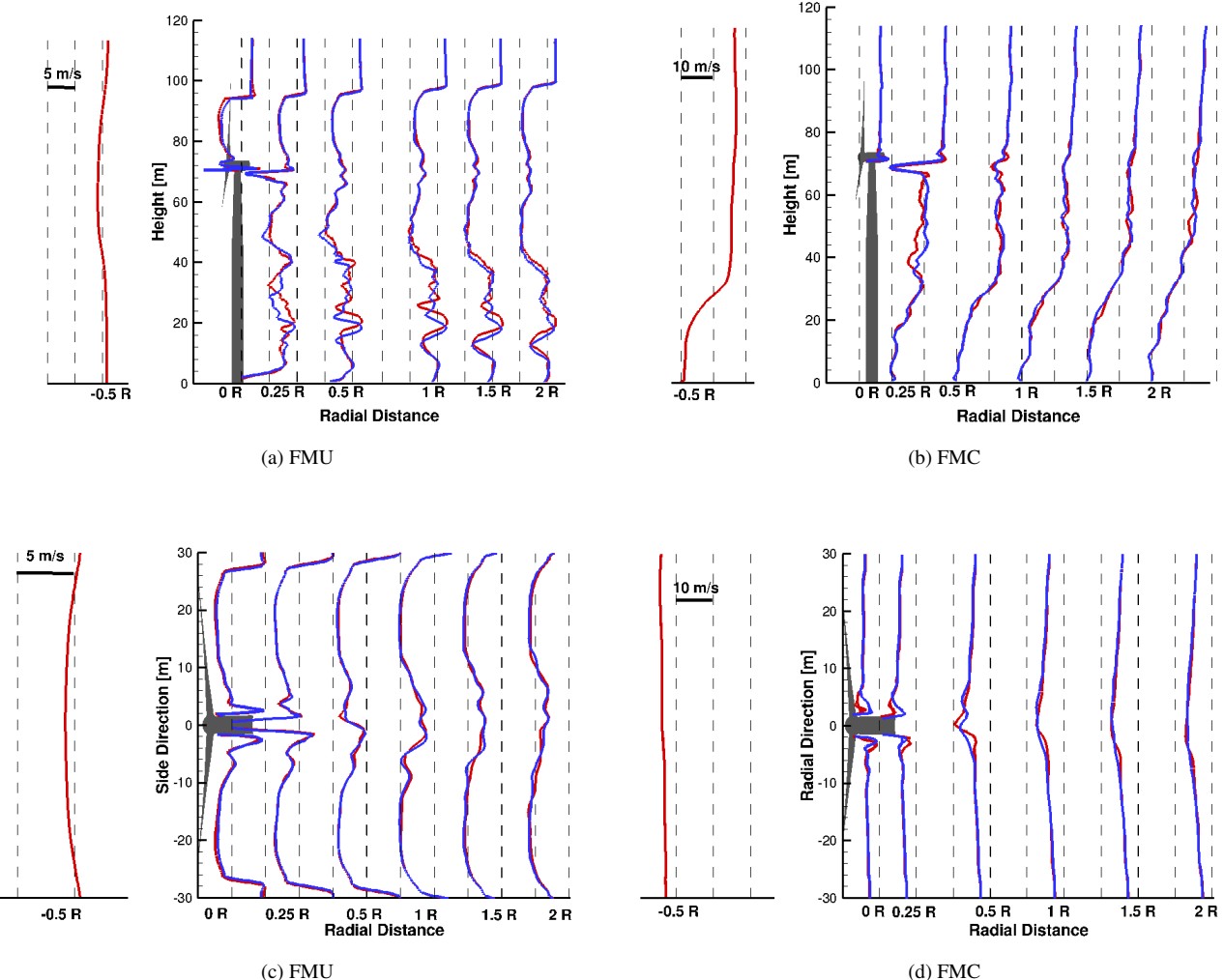

**Figure 10.** Profiles of the velocity component in wind direction at different axial distances from the tower center. x-z (flow direction-height) slices in (a) and (b), x-y (flow-side direction) slices in (c) and (d), including averaged inflow slice. Red lines represent the rigid case, blue lines the beam-coupled case.

## 3.3 Aeroelastic effects on the near wake

In this section the effects of flexibility on the near wake are analyzed. For this reason wind velocity profiles have been extracted at defined radial distances from the hub center and averaged over three revolutions. The rigid and coupled simulations have been run in parallel, so that the same time series could be taken into consideration and averaged. The wind profiles have been extracted for two sections, a x-z (flow direction-height) plane through the tower axis, and a x-y (flow and side direction) plane through the nacelle axis. The cases taken into consideration are FMU and FMC, comparing in both cases the rigid (red lines)

to the beam-based FSI simulation (blue lines). The first has been chosen because is the uniform inflow case and therefore the simplest one, the second one because closest to reality. Because of the higher inflow velocity and the higher pitch in the FMC case, the induction of the rotor decreases. This can be seen in Fig. 10b, where the influence of the rotor presence on the wake shape is negligible. For both FMU and FMC the upper part of the rotor shows to be less influenced by flexibility, where only a small shape discrepancy can be noticed in the closest section to the rotor. On the other side, the lower part of the rotor wake mixes with the tower wake showing different profile shapes up to a distance of 2R in the area between 15 m to 50 m height. The tower is really stiff, and especially in this region its deformation is negligible in comparison to the blade and cannot therefore change the wake shape. This suggests that the discrepancy has to be led back to the blade-tower interaction. In FMU, where the velocity is kept constant, the blade-tower interaction has a strong impact and that is why wake profile differences can be seen up to 2R distances. This effect is lower in 10b because the RPM harmonic peaks of the flapwise blade deformation are not much stronger than the turbulence peaks in as seen in Sec. 3.2.

On the other side, the y-z sections in 10a and 10b are extracted at the hub height and are therefore not influenced by the tower wake. As in the upper part of the rotor in x-z planes, no influence of the wake shape can be noticed independently of the blade pitch angle. Due to the oscillating movement of the nacelle, the hub region shows small discrepancies that disappear already after 2R radial distances.

### 3.4 Difference between flat and complex terrain including flexibility

Both FMT and FMC consider the same turbulent inflow field as input at the inlet, propagating on a flat terrain and up a hill with a forest, respectively. This means also that the time needed by the flow to reach the turbine is not the same between the two cases and therefore differences between flat and complex terrain can be only extrapolated from long simulations in a statistic way. One minute simulations have been computed for all cases, as a compromise between computational time and statistic requirements. On the other side, when the effects of aeroelasticity are of interest, it is possible to compare rigid and coupled simulations timestep per timestep as long as the chosen real time series are coinciding. 11b shows the influence of the forest, leading to really low velocities especially for 40% of the tower. Here a strong cut between a low velocity and a high velocity region occurs, which is completely absent in the case of the flat terrain, which only has shear effects (in addition to turbulence). Additionally, structures of the forest with low velocity and high TI reach the lower half of the rotor as shown in 11c, with direct consequences on loads and deformations.

The effect of the blade deformation on the blade loads is then analyzed for both FMT and FMC. As described in section 2.1.1, those two cases use turbulence with a different shear coefficient and $u_\infty$ to ensure the same hub velocity close to the rotor. The phase averaged thrust force ($F_x$) calculated over 26 revolutions (about one real minute) the rigid cases is shown in Fig. 12a and 12b. 12c and 12d show the respective difference in case of shell-coupled FSI simulations. Loads are normalized according to the respective average value in rigid case. It can be seen that in flat terrain the load distribution is mostly symmetric, with lower loads during the tower passage. The tower passage effect can be seen also in the FMC case, although the load distribution between the upper and lower side of the rotor is not symmetric anymore. This is explained by the retreating blade effect (Letzgus et al., 2021) by which the blade faces an inclined flow due to escarpment of the orography. This is in phase with the

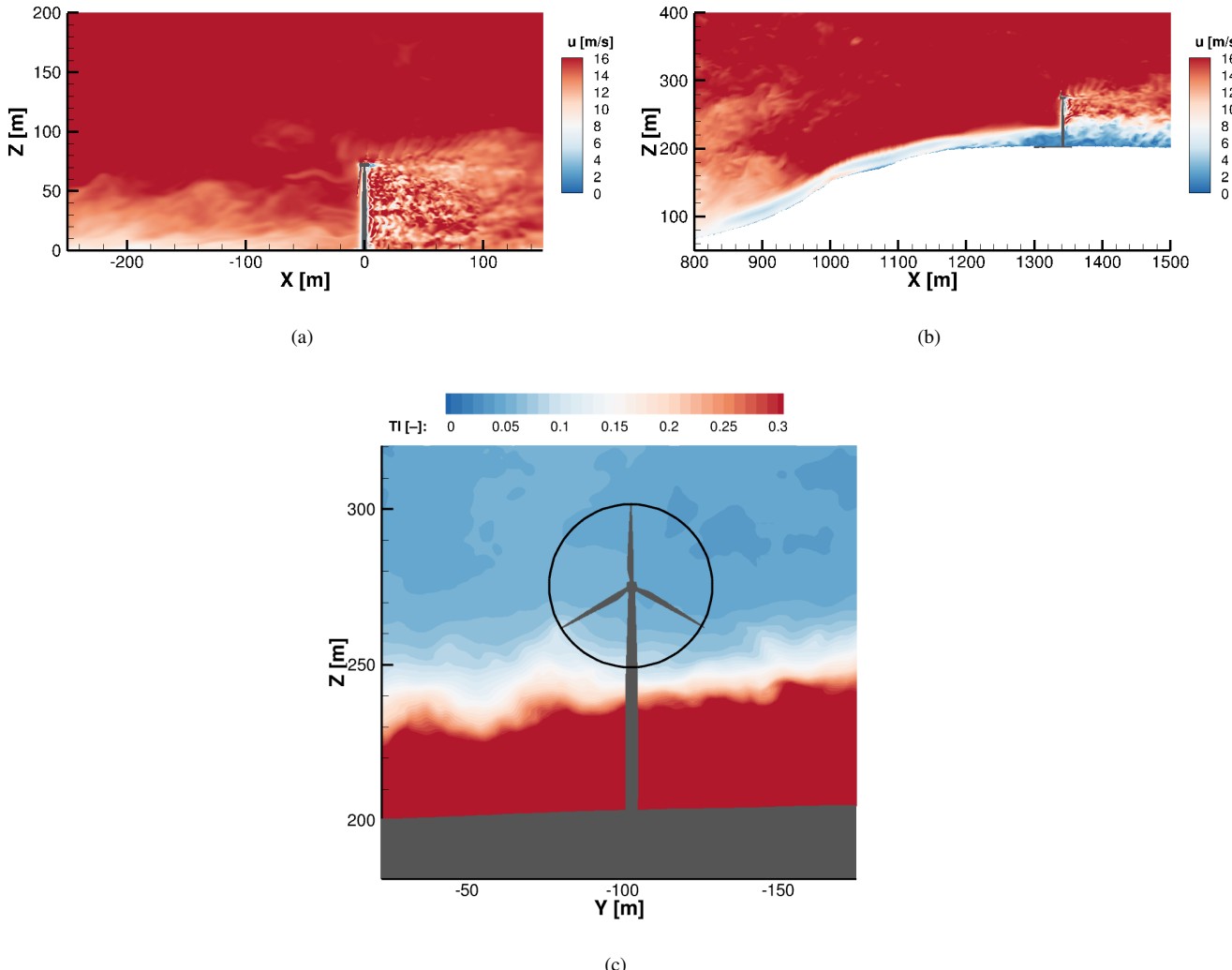

**Figure 11.** Vertical plane with interaction of the turbulent inflow in flat (a) and complex terrain with forest (b). Average TI over one minute in a plane 1R upstream rotor position (c).

blade movement on the Left Hand Side (LHS) of the rotor, and opposite in phase in the Right Hand Side (RHS) of the rotor. This effects leads to a higher AOA on the left-hand side of the plot than on the right-hand side. This phenomenon influences also the tower region, which is not symmetrical around the 180° area anymore. When both cases are considered as flexible, in the flat terrain the largest differences occur at the lower rotor half close to the hub and after the blade-tower passage. Directly after the tower region, i.e. between 180° and 230°, the effect of the blade-tower interaction can be depicted. Due to the inertia

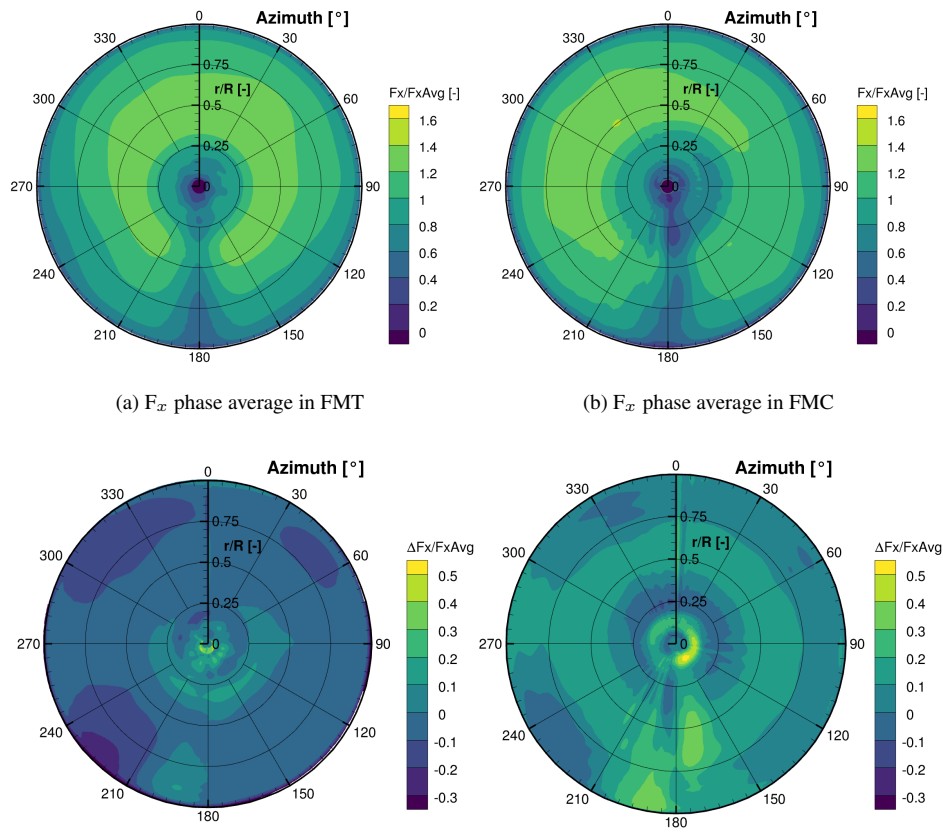

(a) $F_x$ phase average in FMT  (b) $F_x$ phase average in FMC

(c) Difference of $F_x$ between rigid and flexible in FMT (d) Difference of $F_x$ between rigid and flexible in FMC

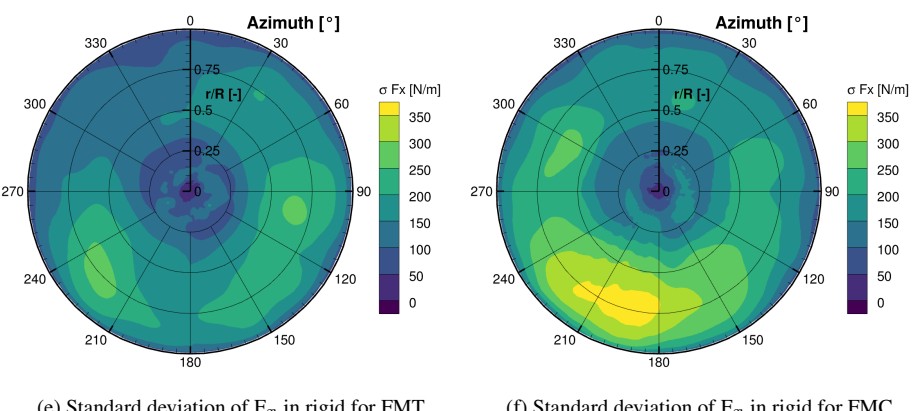

(e) Standard deviation of $F_x$ in rigid for FMT  (f) Standard deviation of $F_x$ in rigid for FMC

**Figure 12.** Phase average of the thrust force of one blade calculated over 26 revolutions. Loads are normalized according to the average in FMT and FMC, respectively. Difference is calculated as ($F_x$ flexible) - ($F_x$ rigid). Standard deviation is calculated for the rigid case.

of the blade, the deformation's peak occurs between 20° after the effective blade-tower passage. Between 180° and 200° the blade faces a lower deformation and moves therefore fast against the wind direction. The deformation velocity of the blade tip reaches up to ±3.5m/s, which represents around 20% of the reference velocity at the hub, with the consequent effect on the relative velocity faced by the blade and therefore loads. The opposite happens in the area between 210° and 240, where the blade tip displacement increases again and moves therefore in the wind direction with a lower relative velocity. The same effect can be observed for the FMC case too (12d), with a further uniform increase of the thrust in the region between 50% and 75% of the blade radius. Additionally from 160° it can be seen that the flexible case develops higher loads. Any connection of this phenomenon with the blade tip deformation due to the blade-tower interaction can be excluded because occurring at least 40° later. The standard deviation of the thrust for the rigid case is shown in Fig. 12e and 12f. Here it can be noticed that in the flat case the lower rotor half shows a higher standard deviation that is due to the presence of shear, while on the other side the case with the complex terrain shows much higher values in the area between 140° and 250°. As shown in Fig. 11c, vortex structures from the forest have a strong influence on the lower rotor half, leading to local low velocities with high turbulence intensities. Summing the retreating blade effect and the forest impact leads to an asymmetrical tower shadow in the rigid case (12b). On the other side, when the blade is bending, the tower shadow impact is more pronounced, becoming again the dominant factor, making the area by 180° more symmetrical again. Additionally, it needs to be considered that the blade presents a cone angle, and therefore a bending deformation increases the rotor disk area, counterbalancing the lower velocities due to the forest wake.

### 3.5 Aeroelastic effects on the Damage Equivalent Loading (DEL)

The DEL is a constant load value that applied over a defined number of cycles, leads to the same damage as a time varying load. In this way it is possible to compare different load signals, and analyze the factors that influence the fatigue on a structure. The approach is based on S-N curves (stress vs number of cycles) of the material on a log-log scale and a rainflow algorithm that recognizes peaks, valleys and therefore fatigue cycles. The formulation used in this study is the one of Hendriks and Bulder (1995), which has been also applied in Guma et al. (2021) on the DANAERO rotor. Considering the conclusions from Sec. 3.2, shell-based FSI simulations have been here used to generate the DEL input. In order to consistently compare the cycle counts, 26 revolutions (around one minute real time) have been taken into consideration. Two different input signals have been chosen, the flapwise ($M_y$) and edgewise ($M_x$) blade root moment respectively. The first one is the also called "bending moment", while the second represents the blade torque. Three cases are compared, FMU, FMT and FMC respectively and the results of the DEL calculations are depicted in Fig. 13. Both signals show similar results, and that is why only the cycle count of the $M_y$ signals has been considered in Fig. 14. The case with uniform inflow is the one showing always the lowest damage level and also the only one in which flexibility is slightly increasing the damage level. Looking at Fig. 14a, it can be seen that flexibility is not only adding small perturbations to the signal (small cycles between 0-10 kNm) but it is also increasing the amplitude of the large cycles (the ones generated by the blade-tower passage). The two cases with turbulence show always larger DEL values compared to uniform inflow (FMU). This is a consequence of the much larger number of small cycles (oscillations due to turbulence) and to the presence of a few very large cycles (up to 500 kNm). The flexibility slightly reduces the DEL values, acting as a positive effect in a similar way for both FMT and FMC. In FMT, cycles with amplitude between 60 and 90 kNm

are damped leading to a higher number of smaller cycles. Similarly in FMC, where very few large cycles up to 420 kNm are damped to smaller cycles. That is why the DEL value slightly decreases, considering that the brighter the cycle, the larger is its influence on the DEL value. It can be concluded that, in case of turbulence where large cycles are detected, flexibility acts as a damper reducing them to a higher number of smaller cycles and therefore smaller fatigue values.

Lastly, a comparison of the DEL for the same FMC case has been performed to confront the beam and shell coupling in Fig. 15. For both inputs, the DEL using the shell model is slightly higher, because of a larger number of smaller cycles, which fits with the oscillations in the deformations shown in Sec. 3.2.

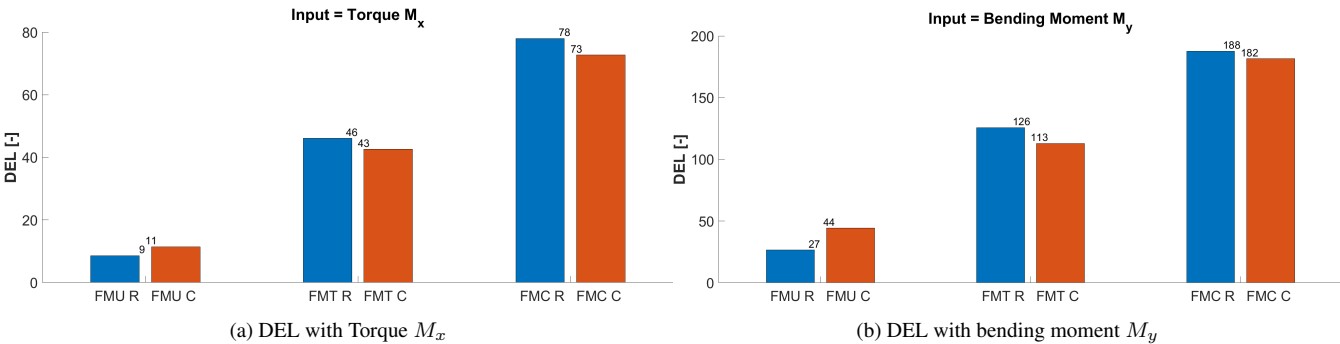

(a) DEL with Torque $M_x$        (b) DEL with bending moment $M_y$

**Figure 13.** Normalized DEL calculation comparison between rigid (blu) and coupled (red) for cases FMU, FMT and FMC.

## 4   Conclusions

In the present work, a high-fidelity Fluid-Structure Interaction coupling between the CFD solver FLOWer and the CSD solver Kratos has been presented. The particular features introduced in this coupling are the possibility to use both beam and shell nonlinear structural elements for the entire turbine and to use both an explicit and implicit coupling algorithm. This coupling has been used to calculate the aeroelastic response and effects of complex terrain on a small sized wind turbine with around 50 m rotor diameter and around 750 kW rated power. This turbine is going to be erected in Stöttener Berg in 2022, a complex terrain with the presence of a hill and a forest. The turbine has been therefore simulated by the use of models increasing their complexity and computational costs from both the aerodynamic, structural and coupling algorithm point of view. A one-third model of the turbine has been utilized to assess the consistency of both structural models, which shows less than 1% difference in case of uniform inflow conditions. The usage of nonlinear models avoided the nonphysical elongation of the blade typical of linear models. A shell model intrinsically considers bend-twist coupling, which is why differences in the predicted torsion occur between the two models, although negligible for such a small blade. The section's shape deformation leads also to negligible differences in the lift calculation, and therefore it has been concluded that in case of single blade calculations in uniform inflow conditions, a beam based coupling is sufficient for such a small turbine. When the aerodynamic complexity increases, i.e. the full turbine in complex terrain is considered, stronger differences occur by adopting either beam or shell structural elements, suggesting that more complicated cases require higher fidelity. Especially in the case that a complex flow

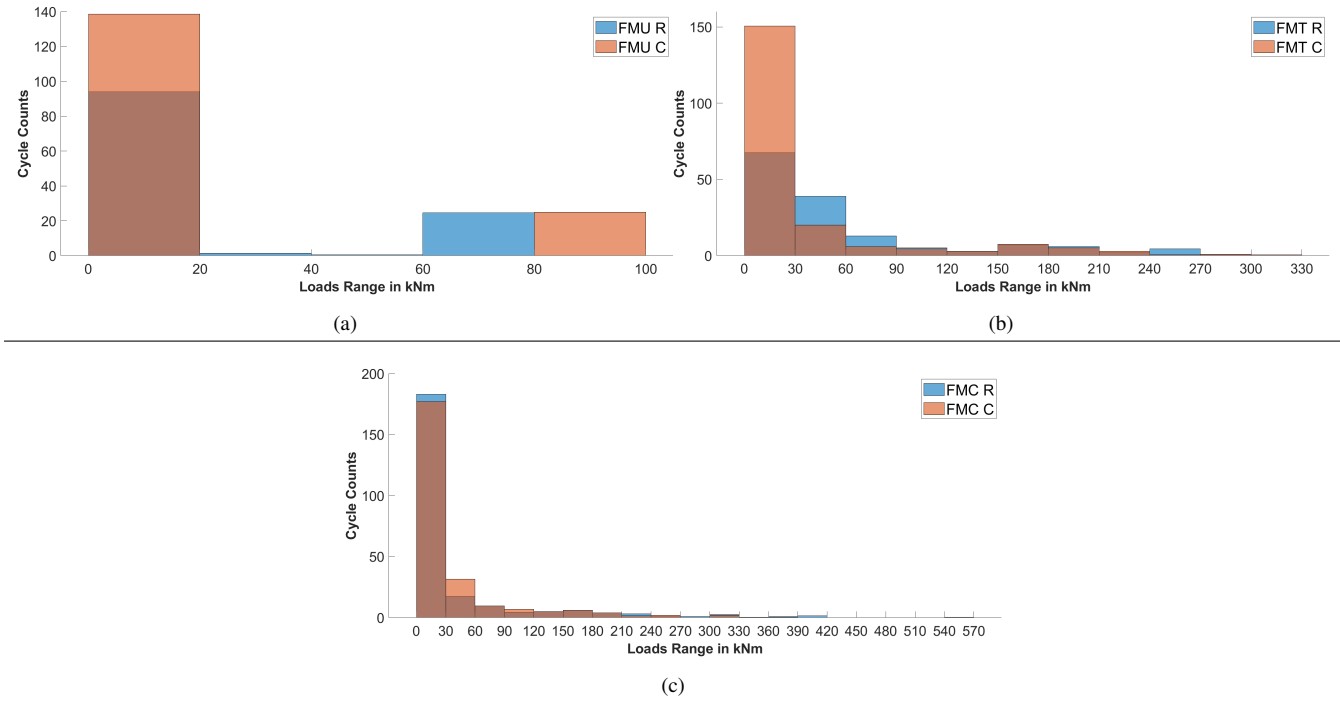

**Figure 14.** Cycle count using the bending moment $M_y$ for the cases FMU, FMT and FMC, both rigid and shell-based coupled.

390 combines with aeroelastic instabilities, the usage of only a beam model, which neglects possible occurring vibrations and the section deformation, can result in misleading results. On the other side it is clear that for only a rough estimate of the loads, the shell model (especially for the entire turbine), is still too expensive from the computational (and pre-processing) point of view. On the other side, the usage of an implicit coupling show no advantage at all in comparison to a much cheaper explicit one. Afterwards, the effects on the near wake by means of velocity profiles' shapes has been analyzed for both the uniform

395 inflow case (with low blade pitch angle) and for the case with complex terrain. Cuts at the hub height show that only the hub region is affected by flexibility in both cases. On the other side, x-z cuts at the tower centre, show that flexibility only affects the velocity profile region behind the tower due to blade-tower interaction blade deformation peaks. These are more evident in case of uniform inflow, while they are hidden by the turbulence frequencies in the complex terrain case, and therefore less evident. Then, the impact of the terrain on aeroelasticity has been discussed by means of phase averaged thrust over one minute

400 simulation. The same time series have been computed for both the rigid and flexible turbine. The blade-tower passage shows here its effect changing the relative velocity between blade and flow and therefore AOA. The flat terrain showed to be mostly affected by the shear while on the other side, the complex terrain is influenced by low velocity vortex structures generated by the forest. Flexibility showed in this case its impact already directly before the tower passage, with an increase of the loads counterbalancing the lower velocities due to the forest wake.

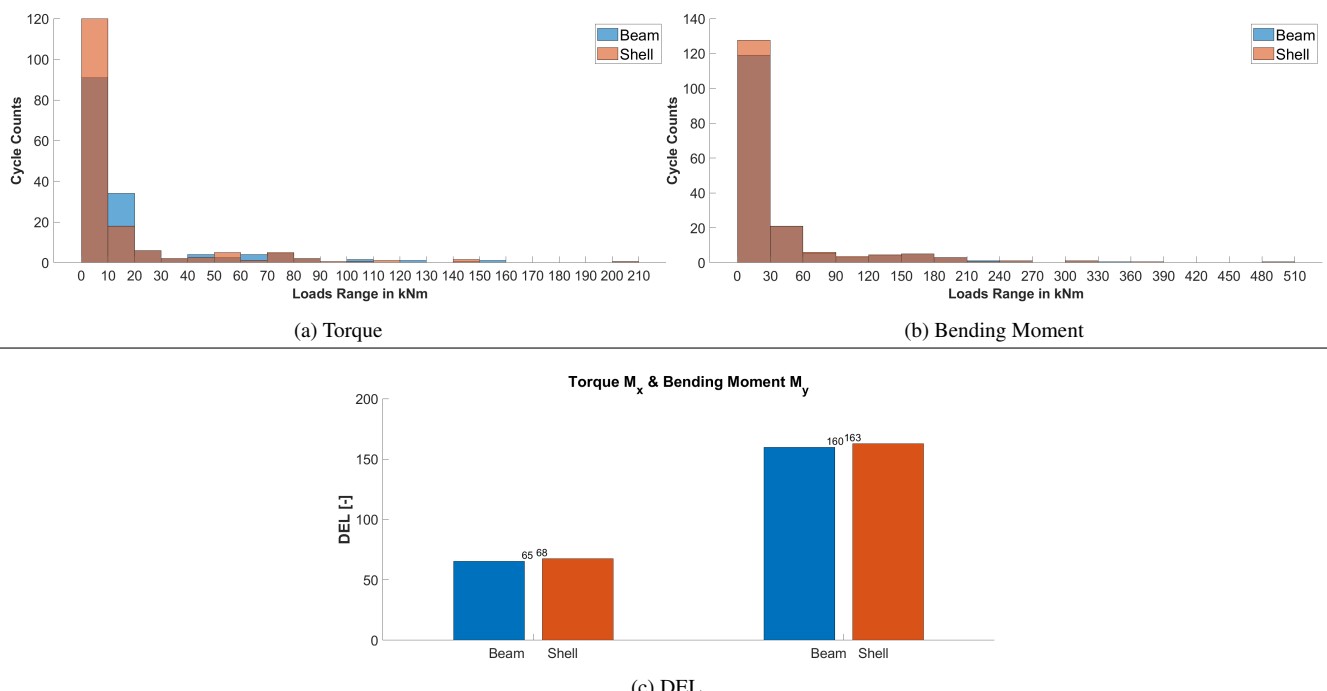

**Figure 15.** Cycle count using the torque (a) and bending moment (b) and resulting DEL (c). Comparison of beam and shell model by FMC.

Finally the effects of aeroelasticity and turbulence are analyzed by means of fatigue calculations (Damage Equivalent Loading). In the uniform inflow case flexibility leads to a higher number of small oscillations and increases the amplitude of the big cycles and therefore yields a higher DEL. On the other side, the cases with turbulence show the opposite effect. This is because in case of turbulent inflow, large cycles are detected and flexibility acts as a damper breaking them into smaller cycles and therefore smaller fatigue values.

The possibility to make in feasible time such high fidelity calculations opens new horizons to further topics such as aeroelastic instabilities, e.g. local buckling analysis and Vortex Induced Vibrations (VIV), for which BEM-beam models still struggle to provide information. Next steps of this work will be the validation of the obtained results with the field data that will be produced after the built of the two research wind turbines in Stöttener Berg. Additionally it is foreseen to introduce a controller, which might have a strong influence on the results especially by high velocities. The way for trustful load prediction for design purposes is still steep, but it is important to take one step at a time. It is the authors' hope, that this work represents one of those steps.

*Data availability.* The raw data of the simulations results can be provided by contacting the corresponding author

*Author contributions.* GG generated the CFD model, developed in collaboration with PB the FSI coupling, run the simulations and made the post-processing. PB generated the CSD model too. PL took care of the complex terrain simulations and generation of inflow data from experimental data. TL and RW supported the research, defined and supervised the work and revised the manuscript.

420

*Competing interests.* The authors declare they have no competing interests

*Acknowledgements.* The authors gratefully acknowledge the founders of the project WINSENT (code number 0324129): the federal Ministry for Economic Affairs and Energy (BMWi) and the Ministry of the Environment, Climate Protection and the Energy Sector Baden Württemberg under the funding number L75 16012. Computer resources were provided by the Gauss Centre for Supercomputing and Leibniz Supercomputing Centre under grant pr94va.

425

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
