# Peer review of "High-fidelity aeroelastic analyses of wind turbines in complex terrain: FSI and aerodynamic modelling"

_Wind Energy Science, 2021_

## Referee Comment (RC1)

**Review of "High-fidelity aeroelastic analyses of wind turbines in complex terrain: FSI and aerodynamic modelling"**

Christian Grinderslev, Postdoc, DTU Wind Energy

**General comments:**

The paper presents results from a set of fluid-structure interaction simulations of a small wind turbine with increasing complexity in both structural representation and wind flow. The most complex case considers a fully resolved wind turbine including tower, set in complex terrain with forest elements upstream creating a highly turbulent inflow. All this with the wind turbine structurally represented as shell FEM elements for the most complex case. The step wise increase of complexity allows the authors to assess the impact of various choices such as the impact of structurally using shell elements rather than beam elements, along with the impact of even considering flexibility in such simulations.
The study concludes that for simple uniform inflow, the choice of structural representation has little impact on the response, whereas a higher dependency is found when considering more complex wind flow. The impact of including flexibility is investigated by looking at velocity deficits in the near wakes along with computations of the damage equivalent loading (DEL) with and without flexibility. It is found that for simple inflow the DEL increases when including flexibility, while the opposite happens in the turbulent inflow.

The paper is well written and easy to follow. The simulations are the most complex I have experienced, considering both high complexity in flow, terrain and structural representation. The step wise increase in complexity allows to easily follow its impacts.

I do have some requests of minor corrections/clarifications as stated below. I especially have many questions to the CFD setups, which could be clarified for the sake of reproducibility. Don't be fooled by the amount of comments. I am very happy about the work, but I think there are low hanging fruits to harvest.

All in all, I recommend publication of the article after the minor adjustments.

A big thanks to the authors for an excellent contribution to the field, showing how far we have come in the world of blade resolved fluid-structure interaction. I'm looking forward to following the continuation of the work and the validation with experiments.

**Specific comments/questions:**

**Section 2.1.1 - CFD setup:**

The **domains** of the various CFD setups are depicted in figure 2.

- It would be nice with an explanation of the abbreviations (FMC, FMT and so on.) I guess FMC is "full model complex" and so on, but it would be nice for the reader to see it more explicitly written out in order to remember it easily later on.

- It would be nice with some dimensions of the domains on the figure, as they seem to be quite different. Also state the position of the turbine in the domain.
- For the FMC case, the domain seems extremely wide, while quite short. First of all, why so wide? Secondly, is the distance from rotor to outlet only 3D as suggested in Figure 11? It might be due to the ratio, that it seems short, but have you conducted a tests of the sensitivity of this? In my experience having the outlet of the domain too close to the turbine can have a big effect on results.

The **grids** are presented here as well:

- How are the cell sizes of the upstream cells which transport the turbulent inflow? What kind of resolution can be captured?
- For the NS-walls at the ground, do you really resolve the boundary layer close to the surface to a y+=1? Or do you use a wall function?
- You write that the sensitivity of the mesh resolution has been investigated in Guma et al, 2018 and Shäffler, 2019. However, in the work by Guma, the flow is simple and steady, while the flow here is much different. The work of Shäffler does not seem to be available but more an internal work?
  Could you state some more about how and to what degree the mesh sensitivity have been investigated? I agree that it's nice that you can use the same mesh for the turbine itself in all the setups, but this might vary from inflow and turbulence model how well suited it is.
- You use a time step of 1 degree rotation. This is quite high compared to my own experience (using another solver), however this of course also depends on the number of sub iterations etc. Did you do a sensitivity study on the time steps? I guess the Chimera grids will jump a few cell sizes for each time step.

**Turbulence** is described in this section as well:

- You write that the FLOWer solver is a URANS/DES solver, however you only state that you use the Shear-Stress-Transport (SST) k-omega model in this work. Does that mean that you use URANS to transport the turbulent fluctuations? I'd think that to be way too dissipative. Could you comment some more on this choice if you really are using URANS?
- How does the turbulence behave over the travelled distance? Does the TI for instance change a lot? And how does the spectrum look?
- You write that you superimpose the turbulence measured from a met mast to the sheared inflow. How did you distribute the turbulence? Did you use tools like PyConTurb to constrain it, or is it more in relation to the spectra and TI you mean it's similar to measurements?
- What is the stratification of the measured/imposed turbulence?
- You describe that you have used different mean velocity and shear for cases FMT and FMC to keep them consistent despite the terrain. However, Table 2 shows the same exponent and velocity for the two cases. Also, you later state that you have ensured the same hub velocity close to the rotor for these two cases, but how large are the differences between bottom and top position of the blade tip if you impose different shears?

**Section 3 - Results**

Your simulations are run for 60 seconds simulation time, which is quite limited, but also understandable with the massive setups that you run here.

- How about transients? Do you for all cases start from a converged state of the stiff rotor?

- Do you turn on the flexibility instantly, and does this give rise to some transient motions, that should have some time to damp out?
  - The reason I ask is for instance Figure 6. Why is the shell response so "wiggly" in such a steady flow? Is this actually physical?

- Any thoughts on the limited time signals, on for instance FFTs and calculations of DEL?
  - Is it enough data to resolve the FFTs sufficiently? Maybe for comparison purposes as they are both the same length.
  - Is it enough data for a good DEL estimate? (I don't have much experience here)

For uniform inflow, you state that the choice of structural representation has very little impact on the response, whereas the shell representation is needed in more complex flows. What is that statement based on? Only the given RMS?

- Would it perhaps be enough to instead use a beam model which could model bend-twist coupling like HAWC2 for instance?
- When I look at the responses shown in figure 8, I agree that there are differences, but I'm not sure I think it justifies the ≈70% increase in computational effort of using a shell model.
- Could you compare the DEL of the two setups (beam vs shell) and see if that justifies the use of more heavy computations?
- In your previous paper on the DanAero turbine, flexibility increased DEL (however much less than the turbulence in itself). What could be the reason for the opposite effect here?

For the near wake investigation, you compare flexible and stiff configurations and see close to no effect.

- It's a little hard to assess how big the differences of deficits really are as there's no scale to compare to.
- Could you add a profile upstream of the turbine? At least for the FMC case it's hard to guess how the incoming flow looks, and therefor hard to evaluate the deficit.

**Minor corrections/suggestions:**

- Avoid abbreviations in the title. FSI sort of excludes the unfamiliar readers.
- Line 43-45. You give the impression that the work by Li, Y et al. was a continuation of Heinz', with the same solvers etc. This is not the case.
- Deformation plots are related to blade radius, but I guess the unit is % not [-]?
- Fig 10, has no quantification on the velocity profiles. Also avoid the use of terms like x-y slices, as the reader is not necessarily familiar with the coordinate system.
- Fig 11. The colorbar makes it look like there is very little shear for FMT, which is not the case. You might reconsider the colorbar limits.
- Fig 14. The figure text is really small.

---

## Referee Comment (RC2)

[referee-annotated manuscript omitted]

---

## Author Comment (AC1)

**Reply to comments by Reviewer Nr. 1**

Giorgia Guma on behalf of the authors
IAG, University of Stuttgart

May 19, 2022

The authors would like to thank the reviewer for his efforts and valuable comments. They are very much appreciated and incorporated into the revised paper.

In the present document the comments given by the 1st reviewer are addressed consecutively. The following formatting is chosen:

- The reviewer comments are marked in blue and italic.

- The reply by the authors is in black color

- A marked-up manuscript is added. Changed section with regard to the comments by reviewer 1 are marked in yellow. Changed sections with regard to comments by both reviewers are marked in gray.

**Specific comments**

1. "*It would be nice with an explanation of the abbreviations (FMC, FMT and so on.) I guess FMC is "full model complex" and so on, but it would be nice for the reader to see it more explicitly written out in order to remember it easily later on*" Thank you very much for your comment. We added its meaning also in Table 1.

2. "*It would be nice with some dimensions of the domains on the figure, as they seem to be quite different. Also state the position of the turbine in the domain.*" Thank you for your comment. For sake of clarity, because Figure 2 is already really full, the authors' decided to introduce a table (Table 2) with the dimensions of the domains from the turbine position.

3. "*For the FMC case, the domain seems extremely wide, while quite short. First of all, why so wide? Secondly, is the distance from rotor to outlet only 3D as suggested in Figure 11? It might be due to the ratio, that it seems short, but have you conducted a tests of the sensitivity of this? In my experience having the outlet of the domain too close to the turbine can have a big effect on results.*" Thank you for your comment. The dimensions of the domains are now explained in table 2. Please, consider that the slices in Fig. 11 are strongly zoomed to see the turbine. As you can see from the table, there is more than enough space for the inflow and the wake development. Regarding the strange aspect ratio for the complex terrain, an explanation for that has been given now in $\boxed{\text{R1:S3}}$ (page 6, line 138).

4. "*How are the cell sizes of the upstream cells which transport the turbulent inflow? What kind of resolution can be captured?*" Thank you for your comment, more information has been addressed in $\boxed{\text{R1:S4}}$ (page 6, line 140)

5. "*For the NS-walls at the ground, do you really resolve the boundary layer close to the surface to a y+=1? Or do you use a wall function?*" Thank you for your comment. No, there is no wall function and the wall surface is indeed solved keeping y+=1.

6. "*You write that the sensitivity of the mesh resolution has been investigated in Guma et al, 2018 and Shäffler, 2019. However, in the work by Guma, the flow is simple and steady, while the flow here is much different. The work of Shäffler does not seem to be available but more an internal work? Could you state some more about how and to what degree the mesh sensitivity have been investigated? I agree that it's nice that you can use the same mesh for the turbine itself in all the setups, but this might vary from inflow and turbulence model how well suited it is.*" Thank you for your comment. Yes, as you presumed, the work from Schäffler was an internal student work. In Guma's work the exactly same mesh has been used, halving and doubling the number of cells. On the other side, in Schäffler work, different mesh properties, characteristics and cells numbers have been tested. Different timesteps and number of iterations as well, although always at rated conditions. In this way it was possible to find the best resolution properties at the surface, trailing and leading edge, wake and pinion area. Nevertheless, in order to be a bit more conservative to address the turbulent case, we decided to use not the "cheapest" one, but the second one. This was then in line with other works at our institute using the same code. As you know from the mexnext project, the mesh requirements are really code dependent. A summary of this explanation has now been addressed in R1:S6 (page 5, line 120).

7. "*You use a time step of 1 degree rotation. This is quite high compared to my own experience (using another solver), however this of course also depends on the number of sub iterations etc. Did you do a sensitivity study on the time steps? I guess the Chimera grids will jump a few cell sizes for each time step.*" Thank you for your comment. The timestep requirements for Chimera are less stringent than the turbulence resolution and coupling stability. Actually, there are some tricks that can be used so that Chimera becomes no issue at all, but this is off topic. The choice of 1 degree comes from a combination of self-made sensitivity studies and expertise at our institute. For example the stiff case in complex terrain has been run with 2 degrees timestep, suggesting the necessity of a stricter one for the flexible case. [Sayed et al.(2016)], who used the previous version of the FSI coupling, also concluded that 1 degree was enough. The number of sub iterations was then chosen to ensure that both beam and shell couplings reach the same residual per timestep as in the stiff case. This has now been addressed in R1:S2 (page 7, line 155).

8. "*You write that the FLOWer solver is a URANS/DES solver, however you only state that you use the Shear-Stress-Transport (SST) k-omega model in this work. Does that mean that you use URANS to transport the turbulent fluctuations? I'd think that to be way too dissipative. Could you comment some more on this choice if you really are using URANS?*" Thank you for your comment. We apologize, we had to specify that DDES is used for the turbulent cases, as now addressed in R1:S7 (page 4, line 113).

9. "*How does the turbulence behave over the travelled distance? Does the TI for instance change a lot? And how does the spectrum look?*" Thank you for your comment. We have chosen the turbulence in the valley so that it matches the measured data at the top of the test site. The turbulence intensity changes slightly due to the speed-up caused by the escarpment. However, on the one hand this was initially taken into account so that the measured and the simulated TI match well and on the other hand the speed-up is lower at these generally high wind speeds than would be the case for rated conditions, for example. In addition, we have chosen a very high spatial resolution and we use a 5th order WENO scheme, so that we can avoid numerical dissipation to a great extent. This has now been addressed in R1:S8 (page 7, line 161).

10. "*You write that you superimpose the turbulence measured from a met mast to the sheared inflow. How did you distribute the turbulence? Did you use tools like PyConTurb to constrain it, or is it more in relation to the spectra and TI you mean it's similar to measurements?*" Thank you for your comment. We selected our turbulence from a 10-minute measurement period of a met mast located near the wind turbine. To realize this, we extracted the spectra and standard deviations of all velocity components of the different measurement positions and generated synthetic turbulence with the Mann model. Our focus was that the turbulence, especially at hub height, matches the measured turbulence very well. We addressed this in $\boxed{\text{R1:S8}}$ (page 7, line 161).

11. "*What is the stratification of the measured/imposed turbulence?*" Thank you for your comment. The atmosphere was neutrally stratified in the observed time period $\boxed{\text{R1:S9}}$ (page 8, line 168).

12. "*You describe that you have used different mean velocity and shear for cases FMT and FMC to keep them consistent despite the terrain. However, Table 2 shows the same exponent and velocity for the two cases. Also, you later state that you have ensured the same hub velocity close to the rotor for these two cases, but how large are the differences between bottom and top position of the blade tip if you impose different shears?*" We have chosen the shear of the velocity profile in flat terrain in a way that it matches to the shear at the wind turbine position in complex terrain. For this purpose, we have evaluated the shear in the calculation in complex terrain and then applied it to the simulation in flat terrain. This means that the slope changes the shear in complex terrain, but at the turbine position the shear is the same as in flat terrain. To avoid misunderstanding, we addressed int he table caption that the values refer to turbine position and tried to explain it better in $\boxed{\text{R1:S8}}$ (page 7, line 161).

13. "*How about transients? Do you for all cases start from a converged state of the stiff rotor?*" Thank you for your comment, and yes all cases restart from converged stiff simulations as now explained in $\boxed{\text{R1:S5}}$ (page 12, line 230).

14. "*Do you turn on the flexibility instantly, and does this give rise to some transient motions, that should have some time to damp out? o The reason I ask is for instance Figure 6. Why is the shell response so "wiggly" in such a steady flow? Is this actually physical?*" Thank you for your comment. The OTM was for us mostly a kind of test case for our coupling, because of its simplicity from both aerodynamic and structural point of view. At that time we introduced only a small initial damping of the deformations, while all full model cases have it for one revolution. So it is in the authors' believe that the "wiggly" aspect would definitely diminish with time, although the edgewise deformations show also in the full cases more perturbations using the shell model.

15. "*Any thoughts on the limited time signals, on for instance FFTs and calculations of DEL? o Is it enough data to resolve the FFTs sufficiently? Maybe for comparison purposes as they are both the same length. o Is it enough data for a good DEL estimate? (I don't have much experience here)*" Thank you for your comment. Of course the longer the better, and indeed, normally DEL calculations should be done with much longer time series, but as you can imagine, we run the simulations for the longest feasible. We focused on the idea that, as long as we compare the same timeseries, although short, the resulting conclusions are meaningful.

16. "*For uniform inflow, you state that the choice of structural representation has very little impact on the response, whereas the shell representation is needed in more complex flows. What is that statement based on? Only the given RMS? - Would it perhaps be enough to instead use a beam model which could model bend-twist coupling like HAWC2 for instance? - When I*"

*look at the responses shown in figure 8, I agree that there are differences, but I'm not sure I think it justifies the 70% increase in computational effort of using a shell model."* Thank you for your comment. Unfortunately we could not compare our shell model to a beam model including bend-twist coupling. Anyhow, even if including it, the profile deformation would not be captured, which can have a larger impact for longer blades. We think the question is what we are looking for. If we only want a rough estimate of the loads, we agree that the usage of a shell model is way too expensive. On the other side, if the goal is to investigate possible instabilities, and possible interaction between vibrations and turbulence, the use of the entire geometry for the structural model can become a key point. We tried to addressed it better in the conclusions.

17. *"Could you compare the DEL of the two setups (beam vs shell) and see if that justifies the use of more heavy computations?"* Thank you for your suggestion, a picture of this, with its comments, have been added, please see R1:S24 (page 21, line 374).

18. *"In your previous paper on the DanAero turbine, flexibility increased DEL (however much less than the turbulence in itself). What could be the reason for the opposite effect here"* Thank you for your comment, this is a really close observation. In the authors' opinion this had to do with the different inflow conditions. The simulations of the DanAero have been performed at the experiment conditions which were with really low inflow velocity (and therefore high induction). The WINSENT rotor was simulated at its rated conditions and then at really high velocity with turbulence. Therefore different results can be expected. On the other side more investigation should be done to confirm these idea, and that is why it has not been mentioned in the paper now.

19. *"For the near wake investigation, you compare flexible and stiff configurations and see close to no effect. It's a little hard to assess how big the differences of deficits really are as there's no scale to compare to."* Thank you for your comment. The objective of the authors was to provide only a qualitative impression of the deficit. But you are right that without a scale is not clear enough. Therefore pictures of the upstream profile with a scale are given.

20. *"Could you add a profile upstream of the turbine? At least for the FMC case it's hard to guess how the incoming flow looks, and therefor hard to evaluate the deficit"* Thank you very much, please see the previous comment.

21. *"Avoid abbreviations in the title. FSI sort of excludes the unfamiliar readers."* Thank you for your comment, we wanted to avoid a too long title, but indeed, it is not clear for everybody.

22. *"Line 43-45. You give the impression that the work by Li, Y et al. was a continuation of Heinz', with the same solvers etc. This is not the case."* Thank you for the comment, I changed it in line R1:S23 (page 2, line 43)

23. *"Deformation plots are related to blade radius, but I guess the unit is % not [-]?"* Yes it should have been in %. We changed it in all deformations plots.

24. *"Fig 10, has no quantification on the velocity profiles. Also avoid the use of terms like x-y slices, as the reader is not necessarily familiar with the coordinate system."* The definitions of the slices have been added in both the text and figure caption. Regarding the first part of your comment, see please see above.

25. *"Fig 11. The colorbar makes it look like there is very little shear for FMT, which is not the case. You might reconsider the colorbar limits"* Thank you very much. We also noticed this, but especially for Figure 11c, this colorbar has been found as best compromise between readability and clearness around the rotor area. We hope it is fine like this.

26. *"Fig 14. The figure text is really small."* Thank you very much, the picture's size has been increased.

**References**

[Sayed et al.(2016)] Sayed, M and Lutz, Th and Krämer, E and Shayegan, Sh and Ghantasala, A and 
[revised manuscript text omitted]

505

---

## Author Comment (AC2)

**Reply to comments by Reviewer Nr. 2**

Giorgia Guma on behalf of the authors
IAG, University of Stuttgart

May 19, 2022

The authors would like to thank the reviewer for his efforts and valuable comments. They are very much appreciated and incorporated into the revised paper.

In the present document the comments given by the 2nd reviewer are addressed consecutively. The following formatting is chosen:

- The reviewer comments are marked in blue and italic.

- The reply by the authors is in black color

- A marked-up manuscript is added. Changed section with regard to the comments by reviewer 2 are marked in orange. Changed sections with regard to comments by both reviewers are marked in gray.

**General comments**

1. "*In my opinion the perspective of the paper in relation to the two structural models employed in the analysis should be changed. It is already important that one can run CFD coupled with a shell type structural model in manageable computer time. Therefore, there is no need for the authors to struggle to point out the advantages of the shell model against the beam model because in this particular configuration and analysis there are no advantages (or at least convincing). The authors first highlight the issue of bend-twist coupling which is well proven that beam models can handle consistently (although the present model does not include this effect), to conclude that the present blade is very stiff in torsion. The same more or less happens with the deformation of the airfoils' shape. It turns out to be negligible. Overall, in the reviewer opinion the shape deformation is far more important difference than bend twist coupling and therefore the assessment study performed is necessary, although it turns out that the effect is negligible for the particular blade. Another point that could be stressed out is that shell models allow for local buckling analyses (identify local buckling modes) while beam models are struggling to provide information about buckling (there are some approximate methods).*"

Thank you very much for your valuable comment. We are really proud we managed to make feasible such complicated simulations, and yes we struggled looking for the advantages of the shell model, because as you also mentioned in the paper, the devil is in the detail. You mentioned the importance that the shell model might have in identifying buckle, and we can say you completely got the point, because these are the studies (not published yet) that we are following at the moment. We decided, to follow your comment to adapt the conclusions in the way to address all these points better. Please see $\boxed{\textbf{R2:G1}}$ (page 23, line 413).

2. "*In the same direction the authors struggle to prove that the difference of the beam against shell predictions is notable in the turbulent wind case. What is clear already from the uniform inflow case is that the two models do not predict the same amount of damping in the edgewise direction. Whether the origin of this difference lies in the structural or the aerodynamic model is not investigated (different structural or aerodynamic damping). With a different damping of the critical edgewise mode, it is reasonable that differences in loads will be magnified when the system is excited by a stochastic inflow*"

Thank you for your comment. As we addressed now additionally in more sections (see R2:S13 (page 12, line 245)) the structural damping has been implemented by TUM exactly in the same way and values in both structural models. That is why, after long discussions within the authors, and with the actual knowledge available from the performed simulations, the reason for the differences has been only given to the absence of some coupling terms in the stiffness matrix of the beam model and the modelling of the entire geometry in the shell. This will probably need further investigations, which are moved now into the conclusion/outlook in R2:G2 (page 23, line 415).

3. "*It should be emphasized that there is still long way to go until load predictions are trustful for design purpose. For example neglecting the controller renders predictions of loads questionable, in particular beyond rated speed*"

Thank you very much, we addressed this again in the conclusions in R2:G2 (page 23, line 415).

**Specific comments**

1. "*1P response of the edgewise loads/deflections is excited by gravitational loads and not by the blade passing in front of the tower. The latter contributes too but definitely much less*"

Thank you this has been addressed in R2:S16 (page 14, line 275).

2. "*At low tip speed ratio values induction is low independent of whether the blade is pitched or not*"

Thank you this has been addressed in R2:S20 (page 15, line 298).

3. "*PSD plots are much easier to read if some filtering is applied in order to better highlight the harmonic and natural frequency peaks. Please consider doing that in the PSD plots of figure 8 and 9. Furthermore it would be nice to introduce grid lines aligned with the harmonics. Then, harmonic peaks will be easier to see*"

Thank you very much for your comment. A Moving-Average has been now applied for figure 8 and addressed in the caption. Because the small amount of data for the explicit-implicit comparison, the usage of a filter on the data showed no advantage for figure 9. The circles on the harmonics are now larger for both pictures, we avoided the grid lines because this is already used to show the eigenfrequency. We hope it is fine like this.

Most comments which were given in the text, are directly edited in the paper in the marked sections.

4. "*I agree on that. It gets more and more important as flapwise deformation increases. In this case 6m flapwise deflection is very high for a 25m blade. So it seems to be a very flexible blade. This is a point that should be highlighted in the discussion and not only inferred by the reader.*"

Please, consider the deformations are shown normalized to the blade radius, so in fig. 6b 6% of 27 meters is meant and not 6 meters. The plots axis and description has been changed for that.

5. "*Then why paying so much attention to this point earlier? Indeed it is an important difference of the models but it should be in advance said that this effect is not important in the specific blade. While slightly more important is the point that follows... about the shape deformation.*"

Thank you very much for your comment. The author's intention is to give always a connection between a result, even if small, and a possible cause for it. As you said, the neglecting of bend-twist coupling showed in this case to have small influence, but it was important for us to check it, and torsion is the most suited parameter for this.

6. "*When it comes to the full wind turbine model do you also model the tower with shells elements? It looks like you do if I believe fig 4 but could you please clarify? If this is the case isn't that a tremendous waste of resources?*"

Thank you for your comment. Some clarifications have been added in $\boxed{\textbf{R2:S14}}$ (page 10, line 193). Yes, in the full model with shell elements also the tower is structurally modeled. As you can see from Table 3, the tower shell model represents around 14% of the total amount of cells. It resulted at the end for the computed cases that it was not necessary, on the other side it was intention of the authors to compare a full beam to a full shell model. Additionally, it is intent of future work to make vibrational studies on the tower, in which the use of a shell model might be decisive.

[revised manuscript text omitted]